# CINEPILE: A LARGE-SCALE VIDEO QUESTION ANSWERING DATASET AND BENCHMARK

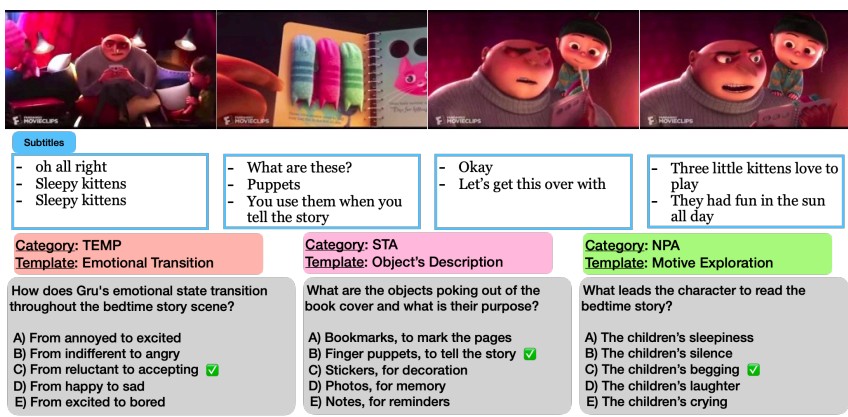

Figure 1: A sample clip (from here) and corresponding MCQs from CinePile.

## ABSTRACT

Current datasets for long-form video understanding often fall short of providing genuine long-form comprehension challenges, as many tasks derived from these datasets can be successfully tackled by analyzing just one or a few random frames from a video. To address this issue, we present a novel dataset and benchmark, CinePile, specifically designed for authentic long-form video understanding. This paper details our innovative approach for creating a question-answer dataset, utilizing advanced LLMs with human-in-the-loop and building upon human-generated raw data. Our comprehensive dataset comprises 305,000 multiple-choice questions (MCQs), covering various visual and multimodal aspects, including temporal comprehension, understanding human-object interactions, and reasoning about events or actions within a scene. Additionally, we fine-tuned open-source Video-LLMs on the training split and evaluated both open-source and proprietary video-centric LLMs on the test split of our dataset. The findings indicate that although current models underperform compared to humans, fine-tuning these models can lead to significant improvements in their performance.

## 1 INTRODUCTION

Large multi-modal models offer the potential to analyze and understand complex videos. However, training and evaluating models on video data presents significant challenges. Most videos contain dialogue and pixel data, and complete scene understanding requires both. Furthermore, existing vision-language models are pre-trained primarily on still frames, while understanding videos requires the ability to identify interactions and plot progressions in the temporal dimension.

In this paper, we introduce CinePile, a large-scale dataset consisting of $\sim$ 305k question-answer pairs from 9396 videos, split into train and test sets. Our dataset emphasizes question diversity, and topics span temporal understanding, perceptual analysis, complex reasoning, and more. It also emphasizes question difficulty, with humans exceeding the best commercial vision/omni models by approximately 25%, and open source video understanding models by 37%.

We present a scene and a few question-answer pairs from our dataset in Figure 1. Consider the first question, `How does Gru's emotional state transition throughout the scene?` For a model to answer this correctly, it needs to understand both the visual and temporal aspects, and even reason about the plot progression of the scene. To answer the second question, `What are the objects poking out of the book cover and what is their purpose`, the model must localize an object in time and space, and use its world knowledge to reason about their purpose.

CinePile addresses several weaknesses of existing video understanding datasets:

**a) Scale:** Its large size allows it to serve both as an instruction-tuning dataset and an evaluation benchmark. When fine-tuned on CinePile's training split, Video-LLaVA achieves a 71% performance gain, demonstrating how large-scale instruction tuning can help bridge the gap between open-source and commercial video understanding models. This scale is enabled by our novel pipeline for automated question generation and verification using large language models. Our method leverages large sets of existing audio descriptions created to assist the visually impaired, which we transcribe and align with publicly available movie clips from YouTube. Using these detailed human scene descriptions, powerful LLMs generate challenging questions without explicit video input.

**b) Generalizability:** Beyond the dataset, CinePile's key strength is its fully automated question generation and verification pipeline, designed for broad applicability. To demonstrate its generalization capabilities, we also generate QAs for longer videos (up to 30 minutes) across diverse domains beyond movie clips. Even with minimal prompt adjustments, our pipeline consistently produces high-quality questions, highlighting its adaptability.

**c) Diversity:** Unlike existing datasets, CinePile does not over-emphasize purely visual questions (e.g., `What color is the car?`) or classification questions (e.g., `What genre is the video?`) that do not require temporal understanding. Rather, CinePile is comprehensive with diverse questions about vision, temporal, and narrative reasoning, including a breakdown of question types to help developers identify blind spots in their models. We propose quantitative metrics to measure semantic diversity of the generated QAs and find that CinePile's diversity is comparable to or greater than that of other datasets, including those curated entirely by humans.

At test time, video-centric models must answer questions using only the dialogue and raw video, without access to the hand-written descriptions used to create those questions. We conduct comprehensive analysis on 24 open- and closed-source models, i.e., include uncovering reasons for the limitations of open-source models, examining the impact of frame rate on performance, and evaluating model accuracy on a "hard" data split. Since its release, CinePile has been adopted by several next-generation Video-LLMs for training and benchmarking, validating its effectiveness as both an instruction-tuning and an evaluation resource. For reviewers' reference, we provide our dataset, evaluation code, and the prompts used in our generation pipeline in the supplementary material.

## 2 CREATING A VIDEO UNDERSTANDING BENCHMARK

Our dataset curation process has four primary components 1) Collection of raw video and related data. 2) Generation of question templates. 3) Automated construction of the Q&A dataset using video and templates, and 4) Applying the refinement pipeline to improve/discard malformed Q&As.

### 2.1 DATA COLLECTION AND CONSOLIDATION

We obtain clips from English-language films from the YouTube channel MovieClips , which hosts self-contained clips, each featuring a major plot point, facilitating the creation of a dataset focused on understanding and reasoning. Next, we collected Audio Descriptions from AudioVault.

**Getting visual descriptions of video for free.** Audio descriptions (ADs) are audio tracks for movies that feature a narrator who explains the visual elements crucial to the story during pauses in dialogue. They have been created for many movies to assist the vision impaired. The key distinction between conventional video caption datasets and ADs lies in the contextual nature of the latter. In ADs, humans emphasize the important visual elements in their narrations, unlike other video caption datasets, which tend to be overly descriptive. We use the audio descriptions as a proxy for visual annotation in the videos for our dataset creation.

**Scene localization in AD.** The video clips we have gathered are typically 2-3 minutes long, while ADs cover entire movies. To align them with video, we transcribe the audio from both the movie clip and

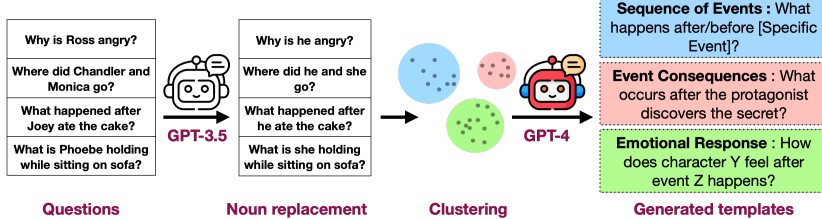

Figure 2: **Question template generation pipeline**: We begin by substituting the first names in human-written source questions and then cluster them. We then feed a selection of questions from each cluster into GPT-4, which outputs "question templates" used in the next stage of dataset creation. See Section 2.2 for more details.

the whole movie AD file using an Automatic Speech Recognition (ASR) system `WhisperX` (Bain et al., 2023), an enhanced version of `Whisper` (Radford et al., 2023) designed to offer quicker inference and more precise word-level timestamps. We then embed the first and last 3 lines of the YouTube clip transcription using a sentence embedding model, `WhereIsAI/UAE-Large-V1`. We similarly embed all the sentences in the movie AD file. We then localize the YouTube clip within the AD file via the rolling window algorithm. We then extract all AD data that lies between the matched start and end of the movie clip embeddings. This localized text contains both the visual elements and the dialogue for the given YouTube clip. This serves as a base text for creating the QA dataset For the rest of the paper, we will refer to the human-written description of the scene as "visual description" and the speaking or dialogue part of the video as "dialogue". When combined, we will refer to both data sources as "**scene-text-annotation**". Since transcriptions don't label sentences as visual descriptions or dialogue, we fine-tuned a BERT-Base model (Devlin et al., 2018) with MAD annotations (Soldan et al., 2022) for classification, achieving 96% accuracy (Appendix C).

## 2.2 AUTOMATED QUESTION TEMPLATES

Many prominent video question-answering benchmarks are created by human annotators, with question-answer pairs curated in two ways: (1) annotators have full freedom to ask questions about a given scene (Tapaswi et al., 2016), or (2) they focus on specific aspects, guided by training or examples to maintain a consistent style (Xiao et al., 2021; Li et al., 2020; Lei et al., 2018; Patraucean et al., 2024). For example, in the Perception Test Benchmark (Patraucean et al., 2024), annotators emphasize temporal or spatial aspects, while in the Next-QA dataset (Xiao et al., 2021), they focus on temporal and causal action reasoning. During early experiments, we found that giving a range of templates and scene-text-annotation to an LLM helped create more detailed, diverse, and well-formed questions. Thus, we adopted a template-based approach for question generation. Instead of limiting questions to a few hand-curated themes, we propose a pipeline to create templates from human-generated questions (shown in Figure 2).

Our starting point is approximately 30,000 human-curated questions from the MovieQA (Tapaswi et al., 2016), TVQA (Lei et al., 2018), and Perception Test (Patraucean et al., 2024) datasets. We cluster these questions, select a few representatives per cluster, and then use GPT-4 to discern the underlying themes and write a prompt. First, we preprocess the questions by replacing first names and entities with pronouns, as BERT (Reimers and Gurevych, 2019) embeddings over-index on proper nouns, hence the resultant clusters end up with shared names rather than themes. For instance, 'Why is Rachel hiding in the bedroom?' is altered to 'Why is she hiding in the bedroom?'. We used GPT-3.5 to do this replacement, as it handled noun replacement better than many open-source and commercial alternatives. The modified questions are then embedded using `WhereIsAI/UAE-Large-V1`, a semantic textual similarity model which is a top performer on the MTEB leaderboard[1]. When the first names were replaced, we observed significant repetition among questions, which prompted us to duplicate them, ultimately resulting in 17,575 unique questions. We then perform k-means clustering to categorize the questions into distinct clusters. We experimented with different values of $k = 10, 50, 100$. Qualitatively, we found $k = 50$ to be an optimal number of clusters where the clusters are diverse and at the same time clusters are not too specific. For example, we see a 'high-school dance' cluster when $k = 100$, and these questions are merged into an 'event' cluster when we reduce $k$ to 50. The Perception Test questions are less diverse as human annotators were

---

[0]Icons in the figures are sourced from Flaticon.

[1]https://huggingface.co/spaces/mteb/leaderboard

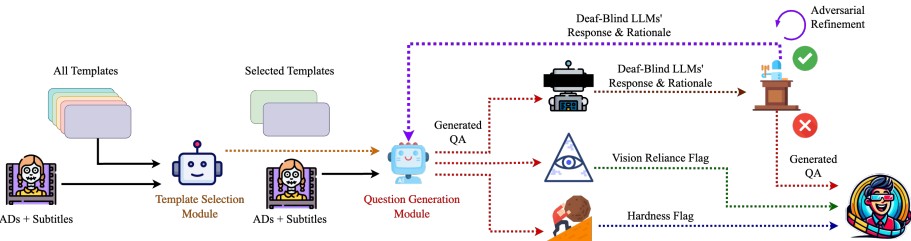

Figure 3: **Question Generation & Quality Checks.** Begins with a set of automated templates and scenes. Filter out the templates relevant to each scene using Gemini. Next, pass these templates along with the annotated-scene-text to GPT-4 to create multiple-choice questions (MCQs). MCQs are then subjected to a refinement pipeline and numerous filters to curate the final dataset. For more details, refer to Section 2.3 and Section 2.4

restricted to creating questions based on a small number of themes, so we used $k = 20$ for this set. The number of questions in each cluster ranges from 60 to 450. We selected 10 random questions from each, and used them to prompt GPT-4 to create relevant question templates (Figure 6 in Appendix D). We did ablations by selecting the closest 10 questions to the cluster center, however observed that random questions produced more general/higher quality templates.

We generate 4 templates for each question cluster, resulting in 300 templates across three datasets. We then manually reviewed all templates, eliminating overly specific ones and merging similar ones. Overly specific templates and their proto-questions looked like "**Pre-wedding Dilemmas:** `What complicates character Z's plans to propose marriage to their partner?`" and "**Crime and Consequence:** `What is the consequence of the character's criminal actions?`". The authors also added many templates complementary to the auto-generated ones, resulting in 86 unique templates. We then manually binned these into five high-level categories: Character and Relationship Dynamics, Narrative and Plot Analysis, Thematic Exploration, Temporal, and Setting and Technical Analysis. For detailed category definitions, template examples, and prototypical questions, see Appendix D & E.

### 2.3 AUTOMATED QA GENERATION WITH LLMS

While the question templates are general, they might not be relevant for a particular movie clip. Hence, we provide Gemini with the scene-text-annotation of a particular scene, asking it to shortlist the 20 most relevant templates, out of which we randomly select 5-6 templates. We then provide a language model with (i) the scene-text-annotation, which includes both visual descriptions and dialogue, (ii) the selected question template names (e.g. 'Physical Possession'), (iii) the prototypical questions for the templates (e.g. "What is [Character Name] holding"), and (iv) a system prompt asking it to write questions about the scene. Through rigorous experimentation, we devised a system prompt that ensures attention to the entire scene, enabling deeper, long-term questions rather than only surface-level ones. We observed that providing the prototypical example prevents GPT-4 from hallucination, and also leads to more plausible MCQ distractors. We also found that asking the model to provide rationale for its answer enhances the quality of the questions. Additionally, we found that including timestamps for the scene-text-annotation augments the quality of generated temporal questions. Through this method, we were able to generate $\approx 32$ questions per video. We analyzed the generated QA pairs and noticed most questions are focused on reasoning or understanding. For diversity, we introduced additional hand-crafted prompt templates for perceptual questions and temporal questions. While GPT-4 performs well across all question templates, Gemini excels particularly with perceptual ones. Therefore, we utilized Gemini to generate a segment of perceptual questions in the dataset, while using GPT-4 for reasoning templates. Our experiments with open-source models indicated subpar question quality, despite extensive prompt tuning. We present qualitative and quantitative investigations into the quality of the generations produced by GPT-4 and Gemini (Appendix F), our QA generation prompt (Appendix N), cost analysis of QA generation (Appendix D).

### 2.4 DATASET QUALITY EVALUATION AND ADVERSARIAL REFINEMENT

While the process above consistently produces well-formed and answerable questions, we observed that some questions are either trivial, with answers embedded within the question itself, or pertaining to basic world concepts that do not require viewing the clip. To identify these, we evaluated our

dataset with the help of a few LLMs on the following axes and we improved the quality of those whenever possible. In the few instances where this was not possible, we removed the questions from the dataset or computed a metric that the users can use in the downstream tasks.

**Degeneracy and educated guessing.** A question is considered degenerate if the answer is implicit in the question itself, e.g., `What is the color of the pink house?`. Similarly, an educated guessing is the most probable answer to the question based on general knowledge, context, or common sense, e.g. `What is the bartender using the shaker for?  a)` **`prepare a cocktail`** `b) do groceries c) collect tips` . Based on an investigation of a subset of the dataset, we found that such questions constituted only a small fraction. However, since manually reviewing all the questions was impractical, we employed three distinct language models (LMs) to identify weak Q&As: Gemini (Anil et al., 2023), GPT-3.5 (Achiam et al., 2023), and Phi-1.5 (Li et al., 2023c). In order to do this, we presented only the questions and answer choices to the models, omitting any context, and calculated the accuracy for each question across multiple models. If multiple models with different pre-training or post-training setups all correctly answer a question, it is likely that the answer was implicit, rather than due to biases of any one.

**Adversarial Refinement.** After identifying weak Q&A pairs, we use an *adversarial refinement* process to repair these Q&A pairs. The goal was to modify the questions and/or answer choices so that a language model could no longer answer them correctly using only implicit clues within the question and answer choices themselves. To achieve this, we used a large language model (LLM), referred to as "deaf-blind LLM", to identify and explain why a question could be answered without extra context. Specifically, when the LLM answered a question correctly, we asked it to provide a rationale for its choice. This rationale helped us detect hidden hints or biases in the question. We then fed this rationale into our question-generation model, instructing it to modify the question and/or answer choices to eliminate these implicit clues. The process repeats until the LLM can no longer answer correctly (adjusting for chance performance), with a maximum of five attempts per question. Given the repetitive and computationally intensive nature of this process, we required a powerful yet accessible LLM that could run locally, avoiding issues with API limits, delays, and costs associated with cloud-based services. As a result, we selected LLaMA 3.1 70B (Dubey et al., 2024), an open-source model that met these desiderata. Through this adversarial refinement process, we successfully corrected approximately 90.94% of the weak Q&A pairs in the training set and 90.24% of the weak Q&A pairs in the test set. Finally, we excluded the unfixable Q&A pairs from the evaluation split (80 Q&A) of our dataset but retained them in the training set (4500 Q&A). We share more details about adversarial refinement in Appendix Sec. P

**Vision Reliance.** When generating the multiple-choice questions (MCQs), we considered the entire scene without differentiating between visual text and dialogue. Consequently, some questions in the dataset might be answerable solely based on dialogue, without the necessity of the video component. For this analysis, we utilized the Gemini model. The model was provided with only the dialogue, excluding any visual descriptions, to assess its performance. If the model correctly answers a question, it is assigned a score of 0 for the visual dependence metric; if it fails, the score is set at 1. In later sections, we present the distribution of the visual dependence scores across different MCQ categories.

**Hardness.** Hardness refers to the inability to answer questions, even when provided with full context used to create them in the first place (i.e., the subtitles & visual descriptions). For this purpose, we selected the Gemini model. Unlike accuracy evaluation, which uses only video frames and dialogues, the hardness metric includes visual descriptions as part of the context given to the model. The authors reviewed all questions flagged as "hard" for verification and corrected any minor issues.

In addition, the authors went through the question in the evaluation split across multiple iterations, and fixed any systemic errors that arose in the pipeline. Furthermore, we conducted a human study to identify potential weaknesses, and we discuss our findings in Section J.

## 3 A LOOK AT THE DATASET

In the initial phase of our dataset collection, we collected ~15,000 movie clips from channels like MovieClips on YouTube. We filtered out clips that did not have corresponding recordings from Audiovault, as our question generation methodology relies on the integration of visual and auditory cues—interleaved dialogues and descriptive audio—to construct meaningful questions. We also excluded clips with low alignment scores when comparing the YouTube clip's transcription with the localized scene's transcription in the Audio Description (AD) file as discussed in Section 2.1. This

Table 1: We compare our dataset, CinePile, to existing video-QA datasets. It is large and diverse. QA types: TP (Temporal), AT (Attribute), NR (Narrative), TH (Theme). '*' denotes automatic annotation, '†' indicates human+automatic annotation, and the rest are fully human-annotated.

| Dataset | Num QA | Avg sec | QA Types | | | |
|---|---|---|---|---|---|---|
| | | | TP | AT | NR | TH |
| *TGIF-QA | 165,165 | 3 | ✓ | ✗ | ✗ | ✗ |
| *MSRVTT-QA | 243,690 | 15 | ✗ | ✓ | ✗ | ✗ |
| How2QA | 44,007 | 60 | ✓ | ✓ | ✗ | ✗ |
| NExT-QA | 52,044 | 44 | ✓ | ✓ | ✗ | ✗ |
| *EgoSchema | 5,000 | 180 | ✓ | ✓ | ✓ | ✗ |
| MovieQA | 6,462 | 203 | ✓ | ✓ | ✓ | ✗ |
| TVQA | 152,545 | 76 | ✓ | ✓ | ✓ | ✗ |
| Perception Test | 44,000 | 23 | ✓ | ✓ | ✗ | ✗ |
| MoVQA | 21,953 | 992 | ✓ | ✓ | ✓ | ✗ |
| IntentQA | 16,297 | UNK | ✓ | ✗ | ✗ | ✗ |
| Video-MME | 2,700 | 1017.9 | ✓ | ✓ | ✓ | ✗ |
| *MVBench | 4,000 | 16 | ✓ | ✓ | ✗ | ✗ |
| †Video-Bench | 17,036 | 56 | ✓ | ✓ | ✗ | ✗ |
| LVBench | 1,549 | 4,101 | ✓ | ✓ | ✓ | ✗ |
| †**CinePile (Ours)** | 303,828 | 160 | ✓ | ✓ | ✓ | ✓ |

process resulted in a refined dataset of 9396 movie clips. The **average video length in our dataset is ~160 sec**, significantly longer than many other VideoQA datasets and benchmarks. We split 9396 videos into train and test splits of 9248 and 148 videos each. We made sure both the splits and the sampling preserved the dataset's diversity in terms of movie genres and release years. We follow the question-answer generation and filtering pipeline which was thoroughly outlined in Section 2. We ended up with **298,887 training points and 4,941 test-set points** with around 32 questions per video scene. Each MCQ includes a question, an answer, and four distractors. As a post hoc step, we randomized the correct answer's position to eliminate positional bias. While we removed degenerate questions from the test split, we retained them in the training set, as they are harmless and might help smaller models develop useful biases found in larger multimodal models like Gemini.

Our dataset's diversity stems from the wide variety of movie clips and different prompting strategies for generating diverse question types. Each strategy zeroes in on particular aspects of the movie content. We present a scene and example MCQs from different question templates in Fig. 1, and many more in the Appendix A. A significant portion of the questions falls under "Character Relationship Dynamics" (41%). This is attributed to the fact that a large number of our automated question templates, which were derived from human-written questions belonged to this category. This is followed by "Setting and Technical Analysis" questions (30.9%), which predominantly require visual interpretation. Regarding vision reliance, As anticipated, questions in the "Setting and Technical Analysis" category exhibit the highest dependency on visual elements, followed by those in "Character Relationship Dynamics", and "Temporal" categories. For the hardness metric, the "Temporal" category contains the most challenging questions, with "Thematic Exploration" following closely behind. Due to space constraints, a detailed visual breakdown is provided in Fig. 8. Finally, we compare our dataset with other existing datasets in this field in Table 1, showing its superiority in both the number of questions and average video length. We provide a more comprehensive comparison in Appendix B due to space constraints. Additionally, we share details, such as the distribution of answer-choice markers, answer-distractor length, for the final dataset in Appendix Q.1.

## 4 MODEL EVALUATION

In this section, we discuss the evaluations of various closed and open-source video LLMs on our dataset, some challenges, and model performance trends. Given that our dataset consists of multiple-choice question answers (MCQs), we assess a model's performance by its ability to accurately select the correct answer from a set of options containing one correct answer and four distractors. A key challenge in this process is reliably parsing the model's response to extract its chosen answer and map it to one of the predefined choices. Model responses may vary in format, including additional markers or a combination of the option letter and corresponding text. Such variations necessitate a robust post-processing step to accurately extract and match the model's response to the correct option. To address these variations, we employ a two-stage evaluation method. First, a normalization function parses the model's response, extracting the option letter (A-E) and any accompanying text.

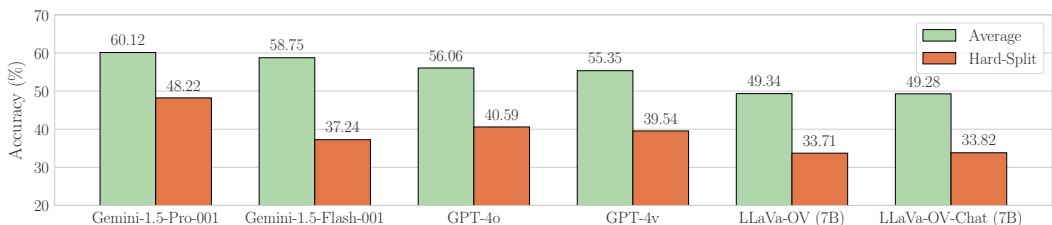

Figure 4: Models' performance on CinePile test split, all questions vs hard questions.

This handles various formats, ensuring accurate identification. The second stage involves comparing the normalized response with the answer key, checking for both the option letter and text. If both match, a score of one is awarded; However, if only the option letter or text appears, the comparison is limited to the relevant part, and the score is assigned accordingly.

Table 2: **Model Evaluations.** We present the accuracy of various video LLMs on the CinePile's test split. We also present Human performance for comparison. We ablate the accuracies across the question categories: TEMP - Temporal, CRD - Character and Relationship Dynamics, NPA - Narrative and Plot Analysis, STA - Setting and Technical Analysis, TH - Thematic Exploration.

| Model | Avg | CRD | NPA | STA | TEMP | TH |
|---|---|---|---|---|---|---|
| Human | 73.21 | 82.92 | 75.00 | **73.00** | 75.52 | 64.93 |
| Human (authors) | **86.00** | **92.00** | **87.5** | 71.20 | **100** | **75.00** |
| Gemini 1.5 Pro–001 | 60.12 | 63.90 | 70.44 | 57.85 | 46.74 | 59.87 |
| Gemini 1.5 Flash–001 | 58.75 | 62.82 | 69.76 | 55.99 | 44.04 | 62.67 |
| GPT-4o | 56.06 | 60.93 | 69.33 | 49.48 | 45.78 | 61.05 |
| GPT-4 Vision | 55.35 | 60.20 | 68.47 | 48.63 | 45.78 | 59.47 |
| LLaVA-OV-7B | 49.34 | 52.13 | 59.83 | 46.54 | 37.65 | 58.42 |
| LLaVA-OV Chat-7B | 49.28 | 52.47 | 58.32 | 46.28 | 37.79 | 58.42 |
| MiniCPM-V 2.6 | 46.91 | 50.10 | 54.21 | 44.52 | 35.61 | 54.74 |
| Claude 3 Opus | 45.60 | 48.89 | 57.88 | 40.73 | 37.65 | 47.89 |
| VideoLLaMA2 | 44.57 | 47.44 | 54.64 | 41.91 | 34.30 | 47.37 |
| InternVL2-26B | 43.86 | 47.10 | 56.16 | 39.03 | 34.16 | 52.63 |
| LongVA DPO | 42.78 | 45.84 | 54.21 | 39.16 | 33.43 | 44.74 |
| InternVL-V1.5-25.5B | 41.69 | 45.07 | 51.19 | 38.97 | 30.09 | 45.79 |
| LongVA | 41.04 | 43.28 | 51.84 | 38.45 | 33.58 | 38.42 |
| InternVL2-4B | 39.89 | 42.99 | 47.73 | 36.23 | 32.99 | 41.58 |
| mPLUG-Owl3 | 38.27 | 40.91 | 45.71 | 33.86 | 33.09 | 46.20 |
| LLaVA-OV-0.5B | 33.82 | 35.88 | 39.96 | 31.66 | 27.03 | 38.42 |
| InternVL2-8B | 32.28 | 35.25 | 40.39 | 28.46 | 24.71 | 38.42 |
| InternVL2-2B | 30.34 | 31.91 | 33.26 | 30.35 | 23.26 | 31.58 |
| VideoChat2 | 29.27 | 31.04 | 34.56 | 25.26 | 27.91 | 34.21 |
| Video LLaVA | 25.72 | 26.64 | 32.61 | 23.63 | 23.26 | 24.74 |
| CogVLM2 | 17.16 | 18.33 | 17.06 | 17.23 | 13.08 | 18.95 |
| InternVL2-1B | 15.97 | 17.65 | 19.22 | 13.25 | 12.94 | 22.63 |
| Video-ChatGPT | 15.08 | 17.06 | 16.34 | 15.17 | 7.26 | 18.58 |
| mPLUG-Owl | 13.93 | 16.15 | 13.16 | 13.03 | 10.48 | 11.54 |

We evaluate 24 commercial and open-source LLM models and we present their performance in Table 2. We discuss additional details about the evaluation timelines, model checkpoints, and compute budget in Section H. We also present human numbers (author and non-author) for comparison. This distinction is important because the authors carefully watched the video (go back and rewatch the video if necessary) while answering the questions. This removes the carelessness errors from the human study. While commercial VLMs perform reasonably well, the very best of OSS models lag ∼10% behind the proprietary models. We present a few QA's which humans got wrong and GPT-4 got wrong and the plausible reason for errors in Section J.

**Gemini 1.5 Pro leads overall; LLaVA-OV tops open-source models.** Among the various commercial VLMs analyzed Gemini 1.5 Pro performs the best, and particularly outperforms the GPT-4 models in the "Setting and Technical Analysis" category that is dominated by visually reliant questions focusing on the environmental and surroundings of a movie scene, and its impact on the characters. On the contrary, we note that GPT-4 models offer competitive performance on question categories such as "Narrative and Plot Analysis" that revolve around the core storylines, and interaction between the key characters. It's important to note that Gemini 1.5 Pro is designed to handle long multimodal contexts natively, while GPT-4o and GPT-4V don't yet accept video as input via their APIs. Therefore, we sample 10 frames per video while evaluating them. Gemini 1.5 Flash, a newly released lighter version of Gemini 1.5 Pro, also performs competitively, achieving 58.75% overall accuracy and ranking second in performance. Its competitive edge over the GPT models is owing to the "Setting and Technical Analysis" category, where it performs significantly better. In open-source models, LLaVA-OV (One Vision) ranks as the best, achieving an overall accuracy of 49.34%. More broadly, while the accuracy of open-source models ranges from 49.34% to 13.93%, it's clear that recent models like LLaVA-OV (released August 2024), MiniCPM-V-2.6 (released August 2024), and VideoLLaMa2 (released June 2024) offer competitive performance.

**Performance significantly drops on the "hard-split".** As discussed in Section 2.4, we provide a "hard split" in the test set consisting of particularly challenging questions. In Figure 4, we compare the performance of the top 6 models on both the average and the hard splits of our dataset. We note that while most models suffer a performance decline of 15%-20% on the hard split; however, the relative ranking among the models remains unchanged. Interestingly, Gemini 1.5 Flash suffers a decline of $\approx 21\%$ compared to $13\%$ for Gemini 1.5 Pro, underscoring the particularly severe trade-offs involved in optimizing the models for lightweight performance on more challenging samples.

Figure 5: Performance comparison of Video-LLaVA after fine-tuning on CinePile's training set. 'Average' refers to the aggregate performance, while the remaining labels represent question types.

**Why are (some) OSS models so far behind?** To better understand the poor performance of certain open-source models, we conducted a qualitative analysis of their raw responses (Appendix I). A key issue was their failure to follow instructions, often producing irrelevant or repetitive outputs that obscured the intended answer. To address this, we explored two alternative accuracy metrics: (a) Embedding Similarity Matching–comparing model responses with answer options in a sentence embedding space (Zhang et al., 2019), and (b) GPT-4 as Judge–extracting answers using GPT-4 (Zheng et al., 2023). We find that while these strategies improve performance by 10-15%, such OSS models still lag behind the top performing ones. Please see Appendix I for details.

**CinePile's train-split helps improve performance** We investigate the impact of CinePile 's training split in enhancing the performance of open-source video LLMs. We selected Video-LLaVA as the baseline and fine-tuned it using CinePile 's training data. For efficient training, we load the model using 4-bit quantization. During fine-tuning, we freeze the base model, and conduct training using Low-Rank Adaptation (LoRA) (Hu et al., 2021). We fine-tuned the model for 5 epochs using the AdamW optimizer (Loshchilov and Hutter, 2017), and compare performance of the fine-tuned model against the base model, as shown in 5. Our results indicate that fine-tuning led to an approximate 71% improvement in performance (increasing accuracy from 25.72% to 44.16%), with gains observed consistently across all question subcategories. These results demonstrate the significant utility of CinePile's training split in enhancing model performance.

**Additional Ablations.** We report additional results on the effect of removing video frames on model performance in Appendix M.1, performance on hard-split (for all models) in Appendix M.2, performance on questions generated for longer and videos different from movie clips in Appendix O. Additionally, we also provide a quantitative analysis of question diversity for CinePile compared to other datasets in Appendix Q.2.1 and rank correlation of rankings on CinePile with other datasets in Appendix Q.2.2.

## 5 RELATED WORK

LVU (Wu and Krähenbühl, 2021), despite being one of the early datasets proposed for long video understanding, barely addresses the problem of video understanding as the main tasks addressed in this dataset are year, genre classification or predicting the like ratio for the video. A single frame might suffice to answer the questions and these tasks cannot be considered quite as "understanding" tasks. MovieQA (Tapaswi et al., 2016) is one of the first attempts to create a truly understanding QA dataset, where the questions are based on entire plot the movie but not localized to a single scene. On closer examination, very few questions are vision focused and most of them can be answered just based on dialogue. EgoSchema (Mangalam et al., 2024) is one of the recent benchmarks, focused on video understanding which requires processing long enough segments in the video to be able to answer the questions. However, the videos are based on egocentric videos and hence the questions mostly require perceptual knowledge, rather than multimodal reasoning. Another recent benchmark, Perception Test (Patraucean et al., 2024), focuses on core perception skills, such as memory and abstraction, across various reasoning abilities (e.g., descriptive, predictive, etc) for short-form videos that they collected by first preparing explicit video scripts. The MAD dataset introduced in (Soldan et al., 2022) and expanded in (Han et al., 2023) contains dialogue and visual descriptions for full-length movies and is typically used in scene captioning tasks rather than understanding. Another issue is this dataset does not provide raw visual data, they share only [CLS] token embeddings, which makes it hard to use. TVQA (Lei et al., 2018) is QA dataset based on short 1-min clips from famous TV shows. The annotators are instructed to ask What/How/Why sort of questions combining two or more events in the video. MoVQA (Zhang et al., 2023b) manually curates questions across levels multiple levels—single scene, multiple scenes, full movie— by guiding annotators to develop queries in predefined categories like Information Processing, Temporal Perception, etc. CMD (Bain et al., 2020) proposes a text-to-video retrieval benchmark while VCR (Zellers et al., 2019) introduces a commonsense reasoning benchmark on images taken from movies. Long video understanding datasets, such as EpicKitchens (Damen et al., 2018), tend to concentrate heavily on tasks related to the memory of visual representations, rather than on reasoning skills. More recently, multiple benchmarks focusing on long video understanding have been released, such as Video-MME (Fu et al., 2024), MVBench (Li et al., 2024), and LVBench (Wang et al., 2024), all having videos from multiple domains such as movies, sports, etc. Most of these datasets require significant human effort to generate questions, with costs increasing as you move toward longer videos. Hence, most of them range on a scale of a few thousand question-answer pairs (CinePile ranges 70-75x more). We discuss works utilizing synthetic data for dataset creation in Section B.

CinePile differs from all the above datasets, having longer videos and many questions to capture the perceptual, temporal, and reasoning aspects of a video. And it is truly multimodal where the person has to watch the video as well as dialogues to answer many questions. Unlike the previous datasets with fixed templates, we automated this process on previously human-generated questions, this let us capture many more question categories compared to previous works. Lastly, our approach to dataset generation is scalable, allowing us to fine-tune video models to improve performance. Moreover, CinePile can easily be extended in the future with additional videos, question categories, and more.

## 6 DISCUSSION AND CONCLUSION

In this paper, we introduced CinePile, a unique long video understanding dataset and benchmark, featuring $\sim$ 300k questions in the training set and $\sim$ 5000 in the test split. We detailed a novel method for curating and filtering this dataset, which is both scalable and cost-effective. Additionally, we benchmarked various recent commercial video-centric LLMs and conducted a human study to gauge the achievable performance on this dataset. To our knowledge, CinePile is the only large-scale dataset that focuses on multi-modal understanding, as opposed to the purely visual reasoning addressed in previous datasets. Our fine-tuning experiments demonstrate the quality of our training split. Additionally, we plan to set up a leaderboard for the test set, providing a platform for new video LLMs to assess and benchmark their performance on CinePile.

Despite its strengths, there are still a few areas for improvement in our dataset, such as the incorporation of character grounding in time. While we believe our dataset's quality is comparable to or even better than that of a Mechanical Turk annotator, we acknowledge that a motivated human, given sufficient time, can create more challenging questions than those currently generated by an LLM. Our goal is to narrow this gap in future iterations of CinePile.

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

# CinePile: A Long Video Question Answering Dataset and Benchmark
## Appendix

**Note:** Also included in the supplementary file are: a) the complete code for loading data, running responses, and evaluating accuracy; b) the Hugging Face dataset objects for the training and test splits, c) the code for running adversarial refinement pipeline, and d) questions generated on longer and different videos.

CONTENTS

## A ADDITIONAL MOVIE CLIP & QUESTIONS EXAMPLES

We present a few examples from our dataset in Figures 15a, 15b, 16a, 16b, 17a, 17b, 18a and 18b.

## B ADDITIONAL RELATED WORK

**Synthetic data with human in the loop.** Training models on synthetic data is a popular paradigm in recent times. We have seen many advances in generation as well as usage on synthetic data in recent times, both in vision Wood et al. (2021); Bordes et al. (2024); Tian et al. (2023); Hemmat et al. (2023) and language Taori et al. (2023); Maini et al. (2024); Li et al. (2023c); Yuan et al. (2024); Wei et al. (2023). For instance, Self-Instruct Wang et al. (2022) proposes a pipeline to create an instruction dataset based on a few instruction examples and categories defined by humans. We mainly derived inspiration and the fact that modern LLMs are quite good at understanding long text and creating question-answer pairs. UltraChat Ding et al. (2023) is another synthetic language dataset which is created by using separate LLMs to iteratively generate opening dialogue lines, simulate user queries, and provide responses. This allows constructing large-scale multi-turn dialogue data without directly using existing internet data as prompts. Additionally, Evol-Instruct Xu et al. (2023), automatically generates a diverse corpus of open-domain instructions of varying complexities by prompting an LLM and applying iterative evolution operations like in-depth evolving (adding constraints, deepening, etc.) and in-breadth evolving (generating new instructions). To our knowledge, we are among the first to apply automated template generation and question synthesis techniques to vision and video modalities using LLMs.

## C DETAILS ON SCENE-TEXT CLASSIFICATION

When we transcribe an AD file, the text contains a human's visual descriptions and the movie's dialogue. However, the transcription model does not label whether a given sentence belongs to a visual description or a dialogue. Since we planned to create a few questions solely on the visual components of the video, the distinction is important to us. To categorize each sentence as either visual or dialogue, we fine-tuned a BERT-Base model (Devlin et al., 2018) using annotations from the MAD dataset (Soldan et al., 2022), which contains labels indicating whether a sentence is a dialogue or a visual description. We applied a binary classification head for this task. For training the classification model, we split the MAD dataset annotations into an 80-20 training-evaluation split. The model achieves 96% accuracy on eval split after 3 epoch training. Qualitatively, we observed that the model accurately classifies sentences in the data we curated.

Table 3: We compare our dataset, CinePile against the pre-existing video-QA datasets. Our dataset is both large and diverse. Multimodal refers to whether both the video and audio data is used for question creation and answering. For understanding different QA types, refer to Section 2.3

| Dataset | Annotation | Domain | Num QA | Avg sec | Multimodal | QA Type | | | |
|---|---|---|---|---|---|---|---|---|---|
| | | | | | | Temporal | Attribute | Narrative | Theme |
| TGIF-QA (Jang et al., 2017) | Auto | Tumblr GIFs | 165,165 | 3 | ✗ | ✓ | ✗ | ✗ | ✗ |
| MSRVTT-QA (Xu et al., 2017) | Auto | Multiple | 243,690 | 15 | ✗ | ✗ | ✓ | ✗ | ✗ |
| How2QA (Li et al., 2020) | Human | Instructional Videos | 44,007 | 60 | ✗ | ✓ | ✓ | ✗ | ✗ |
| NExT-QA (Xiao et al., 2021) | Human | Daily Life Videos | 52,044 | 44 | ✗ | ✓ | ✓ | ✗ | ✗ |
| EgoSchema (Mangalam et al., 2024) | Auto | Egocentric | 5,000 | 180 | ✗ | ✓ | ✓ | ✓ | ✗ |
| MovieQA (Tapaswi et al., 2016) | Human | Movies | 6,462 | 203 | ✓ | ✓ | ✓ | ✓ | ✗ |
| TVQA (Lei et al., 2018) | Human | TV Shows | 152,545 | 76 | ✓ | ✓ | ✓ | ✓ | ✗ |
| Perception Test (Patraucean et al., 2024) | Human | Scripted Videos | 44,000 | 23 | ✓ | ✓ | ✓ | ✗ | ✗ |
| MoVQA (Zhang et al., 2023b) | Human | Movies | 21,953 | 992 | ✓ | ✓ | ✓ | ✓ | ✗ |
| IntentQA (Li et al., 2023b) | Human | Daily Life Videos | 16,297 | Unknown | ✓ | ✓ | ✗ | ✗ | ✗ |
| Video-MME (Fu et al., 2024) | Human | Multiple | 2,700 | 1017.9 | ✓ | ✓ | ✓ | ✗ | ✗ |
| MVBench (Li et al., 2024) | Auto | Multiple | 4,000 | 16 | ✓ | ✓ | ✓ | ✗ | ✗ |
| Video-Bench (Ning et al., 2023) | Human + Auto | Multiple | 17,036 | 56 | ✓ | ✓ | ✓ | ✗ | ✗ |
| LVBench (Wang et al., 2024) | Human | Multiple | 1,549 | 4,101 | ✓ | ✓ | ✓ | ✓ | ✗ |
| **CinePile (Ours)** | Human + Auto | Movies | 303,828 | 160 | ✓ | ✓ | ✓ | ✓ | ✓ |

# D ADDITIONAL QA GENERATION DETAILS

In addition to the hand-crafted perceptual templates, we also create long-form question and answers based on a scene's visual summary. To achieve this, we first generate a visual summary of a video clip. Then, we prompt the model to create question-answers solely based on that summary.

We create a pure visual summary of the scene by using a vision LLM, similar to some of the recent works Wang et al. (2023); Zhang et al. (2023a). First, we use a shot detection algorithm to pick the important frames[2], then we annotate each of these frames with Gemini vision API (gemini-pro-vision). We ablated many SOTA open-source vision LLMs such as Llava 1.5-13B Liu et al. (2023), OtterHD Li et al. (2023a), mPlug-Owl Ye et al. (2023b) and MinGPT-4 Zhu et al. (2023), along with Gemini and GPT-4V (GPT-4-1106-vision-preview). While GPT-4V has high fidelity in terms of image captioning, it is quite expensive. Most of the open-source LLM captions are riddled with hallucinations. After qualitatively evaluating across many scenes, we found that Gemini's frame descriptions are reliable and they do not suffer too much from hallucination. Once we have frame-level descriptions, we then pass the concatenated text to Gemini text model gemini-pro and prompt it to produce a short descriptive summary of the whole scene. Even though Gemini's scene visual summary is less likely to have hallucinated elements, we however spotted a few hallucinated sentences. Hence all the MCQs generated using this summary are added only to the training split but not to the eval split.

**Monetary Costs for Question Generation:** We provide a cost estimate of using GPT-4o for generating QA pairs for one particular scene:

- Base prompt (instructions for question-answer generation and templates): 1,167 tokens

- Movie scene (subtitles and visual descriptions): 465 tokens (average; varies across scenes)

- Total Input Tokens per Scene: 1,632 tokens

- Cost per Input Token: $2.50 per 1M tokens

- Input Cost per Scene**: $\frac{1,632}{1,000,000} \times 2.50 = \$0.00408$

- Average output tokens: 1,582 tokens (average; varies across scenes)

- Cost per Output Token: $10.00 per 1M tokens

- Output Cost per Scene: $\frac{1,582}{1,000,000} \times 10.00 = \$0.01582$

- Total Cost per Scene: $\$0.00408 + \$0.01582 = \$0.0199$

---

**Question Template Automation**

Generate a few templates based on the following questions

[Question 1] Where did the couple meet?
[Question 2] What holiday is coming up when they meet?
[Question 3] What made him break with her?

**Response** Based on the nature of these questions, I can identify a few underlying templates that encompass most of them:

[Template 1] Catalytic Actions
[Proto Question 1] What event acts as a catalyst for the character's next major decision?

[Template 2] Setting and Context
[Proto Question 2] Where does this interaction take place, and how does the location impact the conversation?
...

---

Figure 6: **Extracting templates from human-generated questions.** We share 10 questions from each cluster, and prompt an LLM to create a few templates and a prototypical question. See Figure 2 and Section 2.2 for details.

---

[2]https://www.scenedetect.com/

# E QUESTION TEMPLATE CATEGORY DETAILS

**Character and Relationship Dynamics:** This category would include templates that focus on the actions, motivations, and interactions of characters within the movie. It would also cover aspects such as character roles, reactions, decisions, and relationships.

**Narrative and Plot Analysis:** This category would encompass templates that delve into the storyline, plot twists, event sequences, and the overall narrative structure of the movie. It would also include templates that explore the cause-and-effect dynamics within the plot.

**Thematic Exploration:** This category would include templates that focus on the underlying themes, symbols, motifs, and subtext within the movie. It would also cover aspects such as moral dilemmas, emotional responses, and the impact of discoveries.

**Setting and Technical Analysis:** This category would encompass templates that focus on the setting, environment, and technical aspects of the movie. It would include templates that analyze the location of characters and objects, the use of props, the impact of interactions on the environment, and the description and function of objects.

**Temporal:** This category pertains to questions and answers that assess a model's comprehension of a movie clip's temporal aspects, such as the accurate counting of specific actions, the understanding of the sequence of events, etc.

Table 4: Sample templates and prototypical questions from each of the categories

| Category | Question template | Prototypical question |
|---|---|---|
| Character and Relationship Dynamics (CRD) | Interpersonal Dynamics | What changes occur in the relationship between person A and person B following a shared experience or actions? |
| Character and Relationship Dynamics (CRD) | Decision Justification | What reasons did the character give for making their decision? |
| Narrative and Plot Analysis (NPA) | Crisis Event | What major event leads to the character's drastic action? |
| Narrative and Plot Analysis (NPA) | Mysteries Unveiled | What secret does character A reveal about event B? |
| Setting and Technical Analysis (STA) | Physical Possessions | What is [Character Name] holding? |
| Setting and Technical Analysis (STA) | Environmental Details | What does the [setting/location] look like [during/at] [specific time/-place/event]? |
| Temporal (TEMP) | Critical Time-Sensitive Actions | What must [Character] do quickly, and what are the consequences otherwise? |
| Temporal (Temp) | Frequency | How many times does a character attempt [action A]? |
| Thematic Exploration (TH) | Symbolism and Motif Tracking | Are there any symbols or motifs introduced in Scene A that reappear or evolve in Scene B, and what do they signify? |
| Thematic Exploration (TH) | Thematic Parallels | What does the chaos in the scene parallel in terms of the movie's themes? |

# F QA GENERATION BY DIFFERENT MODELS

In this section, we present example question-answer (QA) pairs generated by GPT-4 and Gemini across various question categories in Table 5 and Table 6. As alluded to in the main paper, we note that GPT-4 consistently produces high-quality questions in all categories. In contrast, Gemini works well only for a few select categories, namely, Character Relationships and Interpersonal Dynamics (CDR), and Setting and Technical Analysis (STA). The gap in quality of the QA generated stems not only from the implicitly better and diverse concepts captured by GPT-4, but also from the hallucination tendencies of Gemini. For instance, in Table- 5, Gemini mistakes the dialogue – "Thank you for talking some sense into me, man", between Eddie and his friend as a suggestion for conflict resolution, and forms a narrative question based on it – "How does Eddie resolve his conflict

Table 5: Comparing question-answer pairs generated by GPT-4 with those generated by Gemini, for the movie clip: The Heartbreak Kid (3/9) Movie CLIP - Taking the Plunge (2007) HD. TEMP refers to Temporal. Please refer to Table 4 for other acronyms.

| Category | GPT-4 Generated QA | Gemini Generated QA |
|---|---|---|
| CRD | Question: What is the significant event that Eddie and Lila are celebrating?
- A) Their wedding ✓
- B) Their first date anniversary
- C) Lila's birthday
- D) Their engagement
- E) Eddie's promotion at work | Question: What is Eddie doing at the beginning of the scene?
- A) Dancing with Lila
- B) Giving a speech
- C) Cutting the wedding cake
- D) Kissing Lila ✓
- E) Talking to his friends |
| NPA | Question: What incident leads to the main character's change in attitude towards marriage?
- A) His friend's advice ✓
- B) His mother's arrival
- C) His bride's beauty
- D) His friend's gift
- E) His bride's dress | Question: How does Eddie resolve his conflict with his friend?
- A) He apologizes for his past behavior.
- B) He confronts his friend about their differences.
- C) He ignores his friend and moves on.
- D) He seeks revenge on his friend.
- E) He reconciles with his friend. ✓ |
| TEMP | Question: How long is the couple planning to take off for their road trip?
- A) One week
- B) Four weeks
- C) Five weeks
- D) Two weeks
- E) Three weeks ✓ | Question: What occurs immediately after the wedding ceremony?
- A) The couple kisses.
- B) The guests congratulate the couple.
- C) The bride's mother arrives. ✓
- D) The couple leaves for their honeymoon.
- E) The groom gives a speech. |
| STA | Question: Where is the gift Eddie's friend gives him supposed to end up?
- A) With Uncle Tito ✓
- B) With Lila
- C) With Eddie
- D) With the wedding guests
- E) With Eddie's mom | Question: What is the primary color of Lila's dress in the scene?
- A) Red
- B) Blue
- C) Yellow
- D) Green
- E) White ✓ |
| TH | Question: How does the emotional tone shift from the beginning to the end of the scene?
- A) From excitement to disappointment
- B) From joy to sorrow
- C) From anticipation to regret
- D) From happiness to surprise ✓
- E) From nervousness to relief | Question: What does the chaotic atmosphere at the reception symbolize in relation to the film's themes?
- A) The unpredictability of life ✓
- B) The challenges of marriage
- C) The importance of family
- D) The power of love
- E) The fragility of relationships |

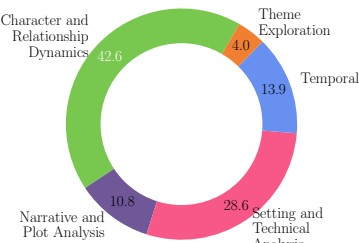

Figure 7: Question category composition in the train split of the dataset.

with his friend?". Similarly, in Table 6, Gemini misremembers the temporal sequence and selects a wrong option as the answer choice for the temporal category. We quantify the quality of generated questions across the different choices of question-generation, and template selection models in Tab. 7. Here, we note that while the GPT-4 & GPT-4 combination results in the fewest degenerate questions, the Gemini & GPT-4 pairing also performs well and is cost-efficient on a large scale.

# G DATASET STATISTICS

We present the question category statistics of the train split in Figure 7 and test-split in Figure 8.

Table 6: Comparing question-answer pairs generated by GPT-4 with those generated by Gemini, for the movie clip: Ghostbusters: Afterlife (2021) - Muncher Attack Scene (3/7) | Movieclips. TEMP refers to Temporal. Please refer to Table 4 for other acronyms.

| Category | GPT-4 Generated QA | Gemini Generated QA |
|---|---|---|
| CRD | Question: How does Phoebe's interaction with the ghost change throughout the scene?
- A) She goes from hiding to attacking. ✓
- B) She goes from attacking to hiding.
- C) She goes from running away to attacking.
- D) She goes from communicating to attacking.
- E) She goes from hiding to running away. | Question: How does the interaction between Phoebe and the ghost change throughout the scene?
- A) Phoebe becomes more aggressive ✓
- B) The ghost becomes more passive
- C) Phoebe becomes more fearful
- D) The ghost becomes more aggressive
- E) They remain neutral towards each other |
| NPA | Question: What major event prompts Phoebe to take drastic action against the ghost?
- A) The ghost scuttles off and pipes burst into flame.
- B) The ghost hides behind machinery.
- C) The ghost starts eating a piece of pipe.
- D) The ghost belches metal fragments that spark and ricochet around them. ✓
- E) The ghost starts searching the ground. | Question: What observation prompts Phoebe to take action?
- A) The ghost's fear of the Aztec death whistle
- B) The ghost's vulnerability to proton blasts
- C) The ghost's search for something on the ground. ✓
- D) The ghost's reaction to Podcast's camera goggles
- E) The ghost's belching of metal fragments |
| TEMP | Question: What happens immediately after the ghost belches metal fragments?
- A) Phoebe ducks down.
- B) The ghost scuttles off and pipes burst into flame.
- C) Podcast blows the Aztec death whistle.
- D) Phoebe powers up and fires a steady stream of protons. ✓
- E) Phoebe pokes her head up. | Question: Between which two events does Phoebe duck down?
- A) The ghost searches the ground and Phoebe pokes her head up.
- B) The ghost chomps on a pipe and Phoebe pokes her head up.
- C) Podcast blows the whistle and the ghost belches metal fragments.
- D) The ghost scuttles off and pipes burst into flame. ✓
- E) Phoebe fires protons and the ghost pokes its head out. |
| STA | Question: Where do Podcast and Phoebe hide during the ghost encounter?
- A) Inside a car
- B) In a building
- C) Behind a tree
- D) Under a table
- E) Behind machinery ✓ | Question: What is the primary material of the object that the ghost is chewing on?
- A) Wood
- B) Metal ✓
- C) Plastic
- D) Rubber
- E) Fabric |
| TH | How does the emotional tone shift throughout this scene?
- A) From calm to chaotic
- B) From fear to courage ✓
- C) From confusion to understanding
- D) From excitement to disappointment
- E) From sadness to joy | Question: How does the emotional tone shift from the characters' initial fear to their determination?
- A) The podcast's calmness inspires Phoebe to become more assertive.
- B) The ghost's search for something on the ground creates a sense of urgency.
- C) The characters' realization that they have a plan instills confidence. ✓
- D) The ghost's belching of metal fragments intensifies the fear and chaos.
- E) The characters' decision to use the trap marks a shift from fear to determination. |

Table 7: Comparison of Template Selection and Question Generation Models in generating better questions (lower degenerate questions) for a subset of movie clips. While the GPT-4 GPT-4 combination performs the best, Template Selection model has minimal effect.

| Template Selection Model | Question Generation Model | % Degenerate Questions |
|---|---|---|
| Gemini | Gemini | 25.12 |
| Gemini | GPT-4 | 18.51 |
| GPT-4 | Gemini | 21.66 |
| GPT-4 | GPT-4 | 13.88 |

# H  ADDITIONAL EVALUATION DETAILS

We use two NVIDIA A40 GPUs, each with 48GB of memory, and two NVIDIA A100, each with memory of 82GB, for experiments with open-source models. The model versions and dates are as follows: Gemini 1.5 Pro [gemini-1.5-pro-001] and Gemini 1.5 Flash [gemini-1.5-flash-001], from May 20th to June 1st, 28th. GPT-4o [gpt-4o-2024-05-13] was used on May 14th, 2024; GPT-4 Vision [gpt-4-turbo], Gemini Pro Vision [gemini-pro-vision], and Claude 3 (Opus) [claude-3-opus-20240229] were used from April 29th to May 10th, 2024. The Gemini 1.5 models throw safety-blocking exceptions for a few of the videos, hence we could only evaluate them on ≈ 4.2k samples out of 4941. The closed-source models in our evaluations (GPT-4, Gemini, Claude families) are released by their respective creators under proprietary licenses. In contrast, open-source

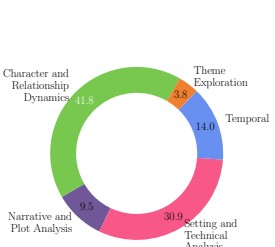 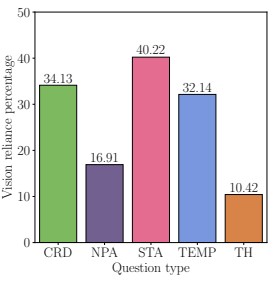 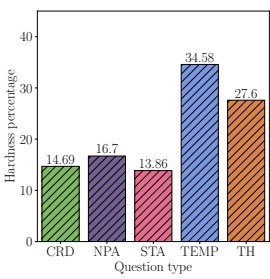

Figure 8: Test split statistics. **Left:** Question category composition in the dataset. **Middle:** Percentage of vision-reliant questions across categories. **Right:** Percentage of hard questions per question category type. TEMP - Temporal, CRD - Character and Relationship Dynamics, NPA - Narrative and Plot Analysis, STA - Setting and Technical Analysis, TH - Thematic Exploration. The colors correspond to the same categories across the plots. Refer to the Appendix for corresponding plots of train split.

models are released under various ope-source licenses such as CC BY-NC-SA 4.0, BSD 3-Clause "New" or "Revised" License, etc.

# I  ADDITIONAL EVALUATION STRATEGIES

Table 8: Analyzing raw responses generated by OSS models, scores assigned by our evaluation pipeline, and corresponding failure modes for the movie clip: Area 51 (2015) - Sneaking Onto the Base Scene (4/10)

| Question | Answer Key | Model Response | Assigned Score | Failure Mode |
|---|---|---|---|---|
| How does the emotional tone transition during the scene?
- A) From despair to hope
- B) From fear to acceptance
- C) From confusion to understanding
- D) From tension to panic
- E) From anxiety to excitement | From anxiety to excitement | "5, 4, 3, 2, 1.\n All right, I think I got it, guys.\n You did it?\n ..... ..... ..... This is the base." | 0 | The model doesn't choose any of the options, rather just repeats all the subtitles. |
| What action does Darren take after observing the broken sensor?
- A) Calls for help
- B) Repairs it
- C) Panics
- D) Ignores it
- E) Suggests next steps | Suggests next steps | "\n ) Darren calls for help." | 0 | While we are able to reliably extract the choice present, it doesn't match the answer key. |
| Where is Jelena in relation to the helicopter at the end of the scene?
- A) Inside
- B) In front
- C) To the side
- D) Directly underneath
- E) On it | Directly underneath | Jelena is standing in front of the helicopter at the end of the scene. | 0 | While we are able to reliably extract the choice present, it doesn't match the answer key. |
| How do Darren and Reed first interact with each other in the scene?
- A) They argue
- B) They fight
- C) They exchange information
- D) They greet each other
- E) They ignore each other | They exchange information | In the scene, Darren and Reed are seen walking together, and they exchange information with each other. | 1 | We are able to extract the response, and it matches the answer key. |

As discussed in Sec. 4 of the main paper, we evaluate a model's performance on CinePile 's test-split by computing its accuracy in choosing the correct answer from a set of multiple-choice options. This involves extracting the chosen answer from the model's raw response and mapping it to one of the predefined answer options. While we perform extensive prompt tuning to ensure the model outputs only the option-letter in its response and rigorously post-process responses to separately extract the chosen option-letter and the corresponding option-text generated (if generated), there remains a

Table 9: Performance of various models on CinePile 's test split, as evaluated using various video captioning metrics – BERTSCoRE (Devlin et al., 2018), CIDEr (Vedantam et al., 2015), ROUGE-L (Lin, 2004).

| Model | BERTScore↑ | CIDEr↑ | ROUGE-L↑ |
|---|---|---|---|
| mPLUG-Owl Ye et al. (2023a) | 0.38 | 0.74 | 0.22 |
| Video-ChatGPT Maaz et al. (2023) | 0.39 | 0.63 | 0.23 |
| Intern-VL-2 (1B) Song et al. (2023) | 0.40 | 1.33 | 0.28 |
| CogVLM-2 Song et al. (2023) | 0.45 | 1.20 | 0.31 |

possibility of errors. The model may not always follow these instructions perfectly and could produce verbose responses with unnecessary text snippets, such as "In my opinion," "The correct answer is," or "... is the correct answer."

Therefore, in this section, we compute traditional video-caption evaluation metrics that emphasize the semantic similarity between the answer key text and the raw model response, instead of exact string matching. We focus our evaluation and discussion on open-source models here, as we qualitatively noted that proprietary models, such as GPT-4V, Gemini-Pro, and Claude, strictly adhere to the prompt instructions, producing only the option letter in their response. Specifically, we calculate the following video-captioning metrics – BERTScore (Zhang et al., 2019), CIDEr (Vedantam et al., 2015), and ROUGE-L (Lin, 2004). BERTScore calculates the contextual similarity between the answer key and model response in the embedding space of a pretrained transformer model like BERT-Base. Calculating the similarity between the latent representations, instead of direct string matching, provides robustness to paraphrasing differences in the answer key and model response. In contrast, CIDEr evaluates the degree to which the model response aligns with the consensus of a set of reference answer keys. In our setup, each question is associated with only one reference answer. The alignment here is computed by measuring the similarity between the non-trivial n-grams present in the model response and the answer key. Finally, ROUGE-L computes the similarity between the answer key and model response based on their longest common subsequence.

We evaluate four open source models, i.e. mPLUG-Owl, Video-ChatGPT, Intern-VL-2 (1B), and CogVLM2, using the aforementioned metrics and report the results in Table 9. In line with the accuracy trend in the main paper. These findings further support the reliability of our normalization and post-processing steps during accuracy computation.

## J   HUMAN STUDY DETAILS

The authors conducted a small human study with 25 graduate student volunteers to evaluate the quality of the CinePile dataset questions. Each participant answered ten randomly sampled multiple-choice questions about two video clips. Our human study survey was granted an exemption by our institute's

Figure 9: Different strategies for evaluating performance on CinePile include: RE (Response Extraction), EBM (Embedding-Based Matching), and GPT-4 Judge (using GPT-4 to assess the raw response).

Institutional Review Board (IRB), and all participants gave their informed consent before viewing the videos and responding to the questions. For full instructions and consent questions given to participants, please refer to Fig. 10-(a). Additionally, we did not collect any personally identifiable information from the participants. It's important to note that our dataset consists of English movies produced in the United States. These films are likely certified by the Motion Picture Association of America (MPAA), which means they adhere to strict content standards and classification guidelines. As a result, they're expected to contain minimal offensive content. An example of the question-answering page can be found in Fig. 10-(b).

Post the study, we interviewed each participant after the survey to ask if they found any systematic issues in any of the questions they were asked to answer about the video. Later, a panel of authors audited all questions where humans got the answer wrong. We noticed that most of the time when a human got a question wrong it was likely due to one of the following reasons (i) due to their inability

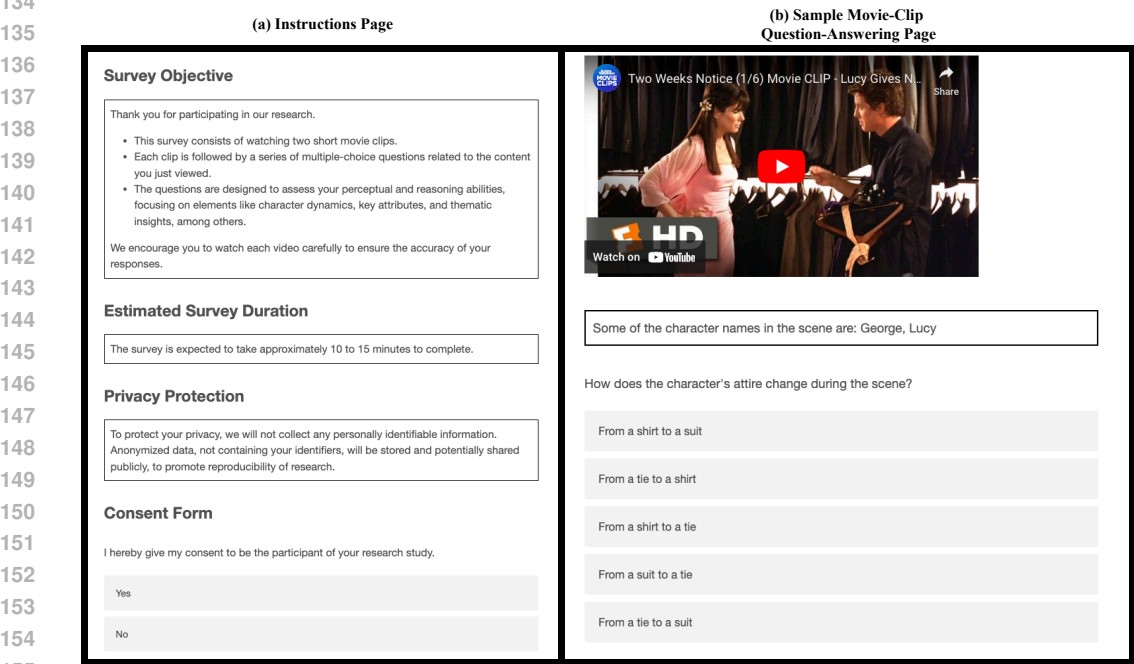

**(a) Instructions Page**

**(b) Sample Movie-Clip
Question-Answering Page**

Figure 10: (*left*) (a) **Instructions Page:** The instructions page at the beginning of the survey, as presented to participants. The participants provide informed consent before viewing any video clip and answering questions. (*right*) (b) **Sample Movie-Clip Question-Answering Page:** An example of one of the movie clips and corresponding question, as presented to the participants. The participants are required to watch the clip and answer the questions by selecting the correct answer choice out of five options.

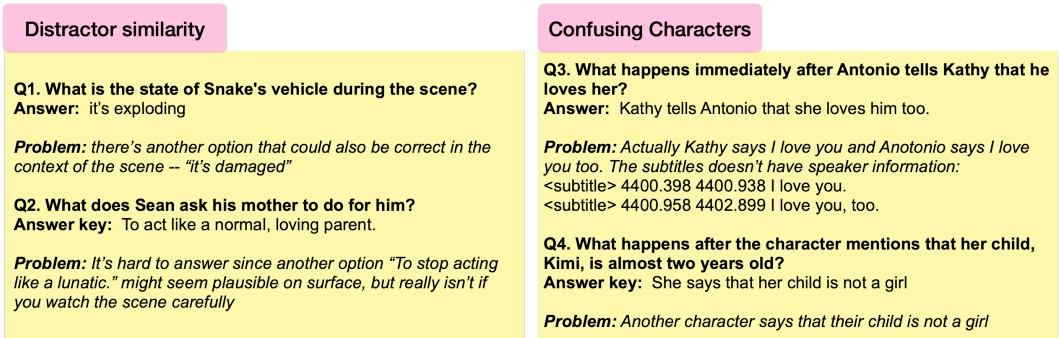

Figure 11: **Sample failure cases from human study**: We conducted a human study to check the quality of questions and we found a few systemic issues. We fixed all systemic issues in the final version of the dataset. The movie clip for Q1 can be found here; for Q2, here; for Q3, here; and for Q4, here.

to attend over the entire clip at once, (ii) due to their inability to understand the dialogue or understand cultural references (iii) carelessness in answering, as the correct answer was indeed present in the video. We did notice some problematic patterns with a small subset of questions. The main issue is distractor similarity, where humans found two plausible answers and they chose one randomly. We present a few such examples in Figure 11. We removed the questions from the test set for which we found ambiguous answers.

We again conducted a second human study on the test set's final version, and the human accuracy is 73%. The authors have independently taken the survey, and the corresponding accuracy is 86%. Once again, a careful investigation by a team of authors indicates that even most of these wrong answers are due to human error and confusion over the many events in a scene. We conclude from this study that many of the questions are answerable but difficult. We present the question category-level performance in Sec. 4 in the main paper.

**Human errors**

**Q1. What is the initial engagement between Sean and his mother in the scene?**
**Answer:** Sean confronts his mother about her past choices
**Participant Response:** Sean asks his mother for help with his college application

*Plausible reason for error: Sean does ask help with college application much later during the scene, maybe the participants have a recency bias, or they didn't pay attention to the operative word "initial" in the question.*

**Q2. What is the first thing Antonio does after revealing the content of the letter from his mother?**
**Answer key:** He hangs his head
**Participant Response:** He gazes out at the water

*Plausible reason for error: For the vast majority of the scene, Antonio is indeed gazing at the water. But after he finishes the relevant content of the letter, the scene cuts to Antonio hanging his head.*

**GPT-4 errors**

**Q3. What is the sequence of events that Antonio narrates to Parker while they sit on the dock?**
**Answer:** Antonio's father told him about a letter, Antonio refused to see it, and then his father threw it away.
**Model Response:** Antonio found a letter from his mother, read it, and then his father threw it away

*Plausible reason for error: The wording of Answer and Model Response may seem the same, but there's key difference that makes the model response incorrect.*

**Q4. What does the chaos caused by the fiery beast parallel in terms of the movie's themes?**
**Answer:** The unpredictability of scientific experiments
**Model Response:** The recklessness of youth

*Plausible reason for error: The model gets influenced by a slightly related scene that talks about being an "adult".*

Figure 12: **Hard questions according to humans and GPT-4 V**: After conducting the human study, we looked at the questions which human got wrong and the questions which GPT-4 got wrong. Some of these questions are difficult and can only be answered by paying careful attention to the video. The movie clip for Q1 can be found here; for Q2 and Q3, here; and for Q4, here.

## K    EXAMPLE DEGENERATE QUESTIONS

Table 10: **Example degenerate questions.** Examples of degenerate questions filtered from CinePile. These questions can be categorized as degenerate for various reasons, including: being answerable through common sense (rows one to three) and the models possibly memorizing the movie scripts (rows four and five)

| Movie Clip | Degenerate Questions |
|---|---|
| Scream (1996) - Wrong Answer Scene (2/12) \| Movieclips | Question: Where does the conversation between the characters take place?
- A) In a restaurant
- B) In a car
- C) In a classroom
- D) At a party
- E) Over the phone ✓ |
| The Godfather: Part 3 (8/10) Movie CLIP - Michael Apologizes to Kay (1990) HD | Question: What thematic element is paralleled in the character's dialogue about his past and his destiny?
- A) The theme of revenge
- B) The theme of fate and free will ✓
- C) The theme of betrayal
- D) The theme of lost innocence
- E) The theme of love and sacrifice |
| The Croods (2013) - Try This On For Size Scene (6/10) \| Movieclips | Question: What happens right before Grug slips on a banana?
- A) Sandy helps Guy hand bananas out to all the monkeys.
- B) The saber-toothed cat roars at them from the bottom of a gorge.
- C) Grug throws a banana down angrily. ✓
- D) Grug puts up his dukes and so does the monkey.
- E) Guy gives Grug a banana. |
| Rugrats in Paris (2000) - We're Going to France! Scene (1/10) \| Movieclips | Question: What event prompts Kira Watanabe to call Mr. Pickles?
- A) The robot's destruction of the village.
- B) The robot's popularity among the villagers.
- C) The malfunction of the giant robot. ✓
- D) The villagers' protest against the robot.
- E) The robot's successful performance. |
| Bottle Rocket (3/8) Movie CLIP - Future Man and Stacy (1996) HD | Question: What happens immediately after Anthony and Dignan finish eating their sandwiches on the patio?
- A) Anthony chews a nut.
- B) A guy in a brown shirt approaches them. ✓
- C) Stacey Sinclair introduces herself.
- D) Anthony tells his story about the beach house.
- E) Anthony goes to clean the pool. |

As discussed in Section 2.4 of the main paper, most question-answers generated are well-formed and include challenging distractors. However, a small minority are degenerate in that they can be answered directly, i.e., without viewing the movie video clip. To automatically filter out such questions, we formulate a degeneracy criterion. If a question can be answered by a wide variety of models without any context—that is, all models select the correct answer merely by processing the question and the five options—we label it as a degenerate question. In this section, we present and discuss some of these degenerate questions in Table 10. We note that a question can be categorized as degenerate due to multiple possible reasons. For instance, consider the questions, "Where does the conversation between the characters take place?", and "What happens right before Grug slips on a banana?". The answer key for these corresponds to the most common-sense response, and the models are able to reliably identify the correct choices ("Over the phone", "Grug angrily throws a banana down") from among the distractions. There's another type of question that models might answer correctly if they've memorized the movie script. For example, the question, "What event prompts Kira Watanabe to call Mr. Pickles?" from the movie Rugrats in Paris, is accurately answered. This likely happens because of the memorization of the script and the distinct character names mentioned in the question.

## L  BROADER IMPACT STATEMENT

We acknowledge the potential for biases inherent in large language models, particularly regarding gender, race, and other demographic factors. Given our use of such models to generate question-answer pairs, there is a risk that these biases may be reflected in the generated content, potentially impacting downstream models trained on this data. While we manually reviewed and filtered problematic questions in the evaluation set, the scale of the training set made it infeasible to apply the same level of scrutiny. Additionally, as most of our movie clips originate from the "global west," there is a possibility that certain stereotypes may be perpetuated. Regarding our human study, we obtained an exemption from our Institute's Review Board (IRB) for the involvement of graduate students. For the dataset release, similar to many existing works (Lei et al., 2018; Tapaswi et al., 2016; Wang et al., 2024; Fu et al., 2024), we release the dataset under the CC-BY-NC-4.0 license, limiting its use to non-commercial, academic purposes. We host the dataset on Hugging Face, requiring users to agree to the license terms before access. Additionally, We do not distribute any raw video content directly; rather, we provide URLs redirecting to YouTube, ensuring compliance with YouTube's Terms of Service (YouTube, 2024).

## M  ADDITIONAL EVALUATION RESULTS

### M.1  FRAME RATE ABLATION

In this section we perform an ablation to investigate the utility of visual frames (from a model's perspective) by completely remove the visual frames and experiment solely with the provided dialogue when evaluating Video-LLMs. We do exactly this in Table 11, and observe that for all models, except Video-ChatGPT, performance significantly declines when evaluated with "only subtitles." This effect is more pronounced in commercial models compared to open-source ones. It appears that better overall models also tend to utilize visual information more effectively. To further investigate the impact of temporal sampling, we also examine model performance when varying the number of sampled frames: [1,8,16,32] on a subset of CinePile questions and plot the results in Fig. 13. Due to the high cost of running these ablations on closed-source models like Gemini, we focused primarily on open-source models from our earlier experiments, adding a new model, MiniCPM-V 2.6. Our findings show that model performance consistently improves as the number of frames increases, except for Video-ChatGPT, which shows no consistent gains. The improvement is proportional to the model's overall ranking in our benchmarks. MiniCPM-V 2.6 shows the most significant performance gains with additional frames, followed by VideoLLaMa2, while Video-ChatGPT's performance remains relatively unchanged, underscoring its limited reliance on visual inputs.

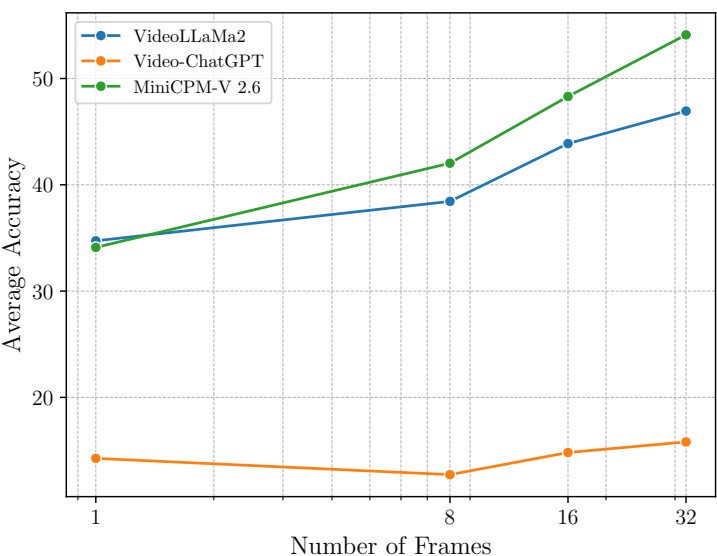

Figure 13: Effect of varying number of samples on overall performance of Video-ChatGPT, VideoLLaMA2, and MiniCPM-V 2.6 on a subset of questions from CinePile.

Table 11: Performance of models with video and subtitles (base case), and when only with subtitles on a subset of CinePile. TEMP - Temporal, CRD - Character and Relationship Dynamics, NPA - Narrative and Plot Analysis, STA - Setting and Technical Analysis, TH - Thematic Exploration.

| Model | Average | CRD | NPA | STA | TEMP | TH |
|---|---|---|---|---|---|---|
| Gemini 1.5 Pro | 51.72 | 51.61 | 56.25 | 55.45 | 40.62 | 50.00 |
| (Only Subtitles) Gemini 1.5 Pro | 34.53 | 35.87 | 44.44 | 31.35 | 32.60 | 36.36 |
| GPT-4o | 50.45 | 51.14 | 66.66 | 52.54 | 34.78 | 45.45 |
| (Only Subtitles) GPT-4o | 37.23 | 45.03 | 44.44 | 29.66 | 28.26 | 45.45 |
| Video-LLaMA2 | 38.44 | 45.80 | 40.74 | 36.44 | 19.56 | 54.54 |
| (Only Subtitles) Video-LLaMA2 | 33.33 | 41.22 | 40.74 | 27.11 | 17.39 | 45.45 |
| Video-ChatGPT | 12.92 | 16.80 | 3.70 | 12.82 | 6.52 | 20.00 |
| (Only Subtitles) Video-ChatGPT | 16.16 | 22.04 | 11.53 | 12.71 | 13.04 | 9.09 |

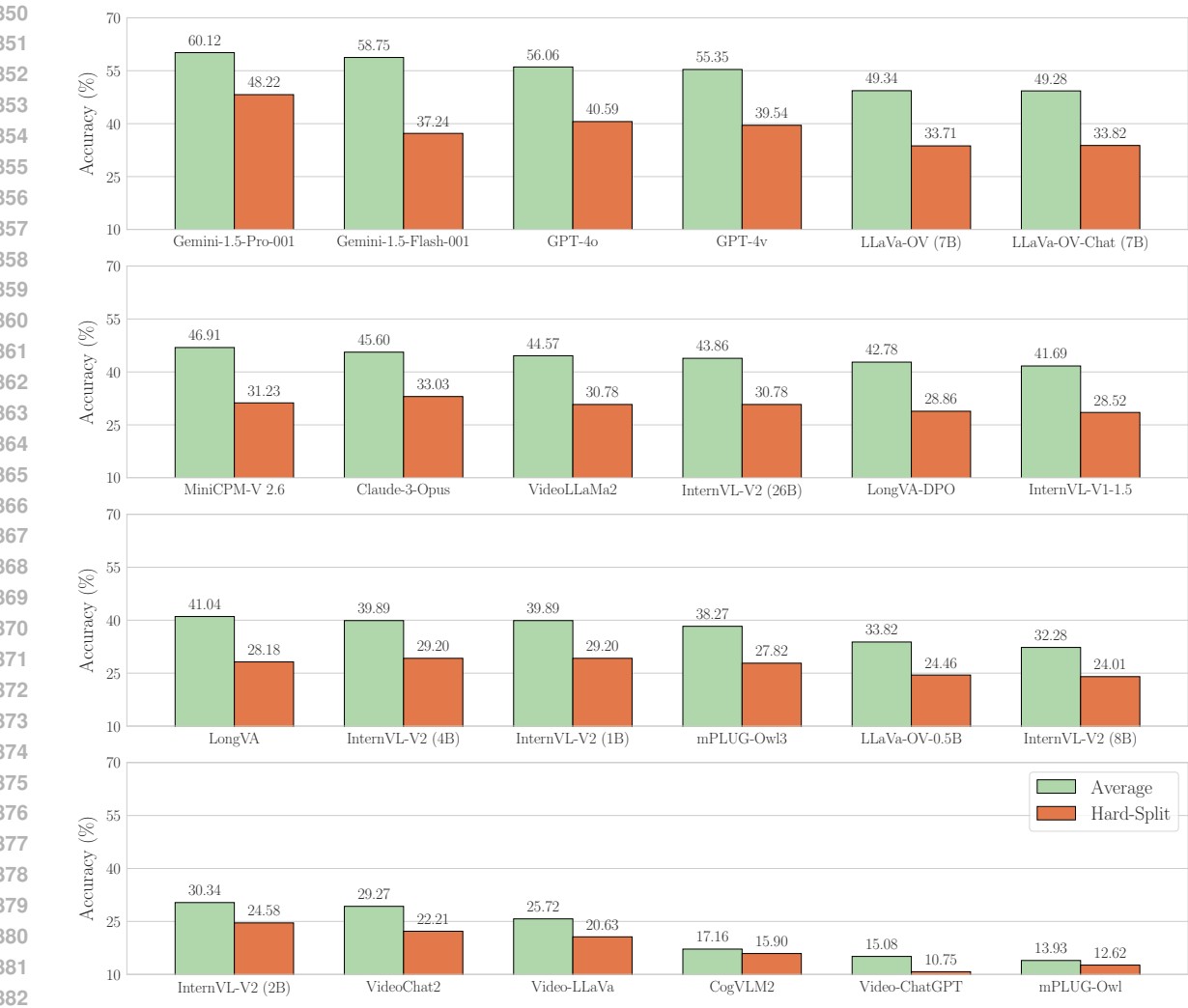

Figure 14: Models' performance on CinePile test split, all questions vs hard questions.

## M.2 PERFORMANCE ON HARD-SPLIT

# N   QA GENERATION PROMPT

As the curator of an advanced cinema analysis quiz, your expertise lies in designing intricate and diverse multiple-choice questions with corresponding answers that span the entire spectrum of film analysis.

- **Objective:** Create diverse and challenging questions based on the film analysis spectrum templates provided below. This spectrum is divided into five subcategories, each comprising several templates. Each template includes a title and a corresponding prototypical question or guideline. Avoid directly replicating the template title and these prototypical questions. Instead, your questions should reflect these elements' essence, even if not explicitly using the category titles in the question's wording.

**Mandatory Guidelines:**
- **Template Use:** Use the provided question templates as a strict guide, ensuring that your questions are both relevant to the scene and varied in their analytical perspective. The prototype question in each template is for inspiration and should not be copied. Your questions should subtly reflect the prototype's essence, tailored to the specifics of the scene.

- **Sub-Category Balance:** Ensure to generate an equal number of questions from each subcategory. This balance is crucial to cover a wide range of analytical perspectives.

- **Question and Answer Format:**

- **Selected Template:** Indicate the film analysis Sub-Category and corresponding template your question is inspired by, without restricting the question's phrasing to the template's title.

- **Questions:** Limited to one or two lines, formulated to be insightful and not overtly indicative of the answer. Avoid using direct template titles or overly descriptive language that could hint at the correct answer.

- **Answers:** Five options per question, formatted as "**- A)**, **- B)**, **- C)**, **- D)**, and **- E)**", concise and reflective of the question's depth.

- **Answer Key:** Specify the correct answer clearly with the formatting, "**Correct Answer:**", in the line following all the answer options.

- **Rationale:** Write a rationale explaining the correctness of the "Answer Key" based on the scene's context in the next line.

**Input Information Format:**

- Movie scene details will be provided in a structured format comprising two distinct categories, and the relevant scene description. The two categories are as follows:

- **<subtitle>** for character dialogues (to be used only for identifying character presence, not actions or dialogue content).

- **<visual descriptions>** for noting characters' presence, attributes, thematic elements, etc., within the scene.

**Movie Scene:** {MOVIE_SCENE_TS}

- **Spectrum of Film Analysis with Templates:**

Sub-Category: Character Analysis
{TEMPLATES_CHAR}

Sub-Category: Narrative Understanding
{TEMPLATES_NARV}

Sub-Category: Scene Setting
{TEMPLATES_SETTING}

Sub-Category: Temporal
{TEMPLATES_TEMPORAL}

Sub-Category: Theme
{TEMPLATES_THEME}

**Instructions:** Your task is to generate clear, unique, and insightful question-answer pairs strictly following the provided templates. Ensure the distribution of questions covers all subcategories evenly. Strictly avoid using words in the questions that give a strong hint about the answer. You can achieve this by keeping the questions concise and not using too many adjectives or adverbs in the question. Incorrect answers must be plausible and closely mirror the correct answer in length and form. The correct answer should not be deducible solely from the question and/or the wrong answers. After presenting all the options, the correct answer must be distinctly specified, but separate from the list of choices. Additionally, provide a concise rationale about why the question-answer falls into one of the selected templates from the Spectrum of Film Analysis by giving verbatim evidence from the subtitles and/or visual descriptions in the movie scene information.

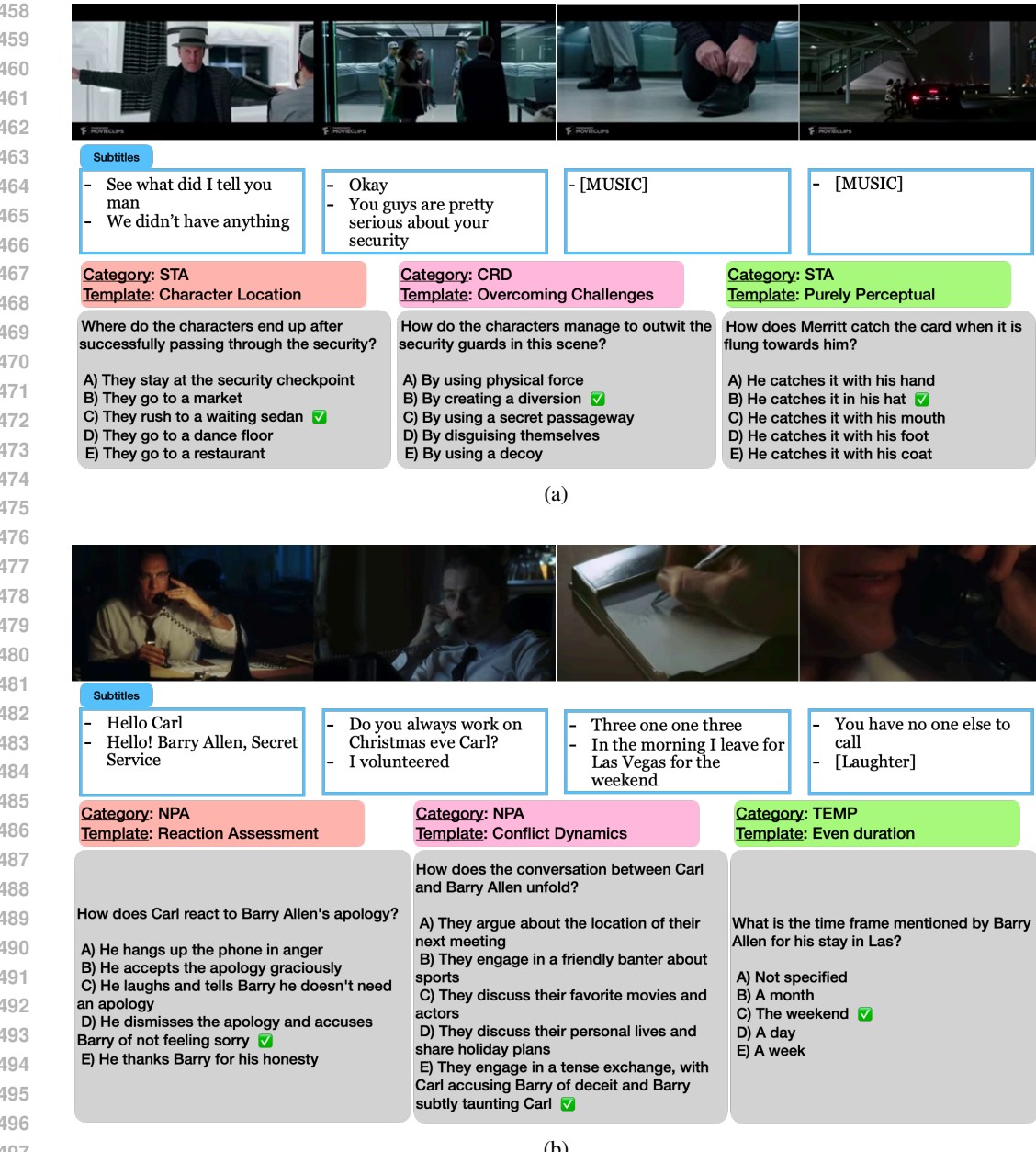

Figure 15: **Example movie clip and multiple-choice questions from CinePile**. The first and second rows depict a selection of image frames extracted from movie clips from (a) Now You See Me 2, and (b) Catch Me if You Can, accompanied by their corresponding subtitles. The next row showcases example questions along with the question template shown in colored headers. TEMP refers to Temporal. Please refer to Table 4 for other category acronyms.

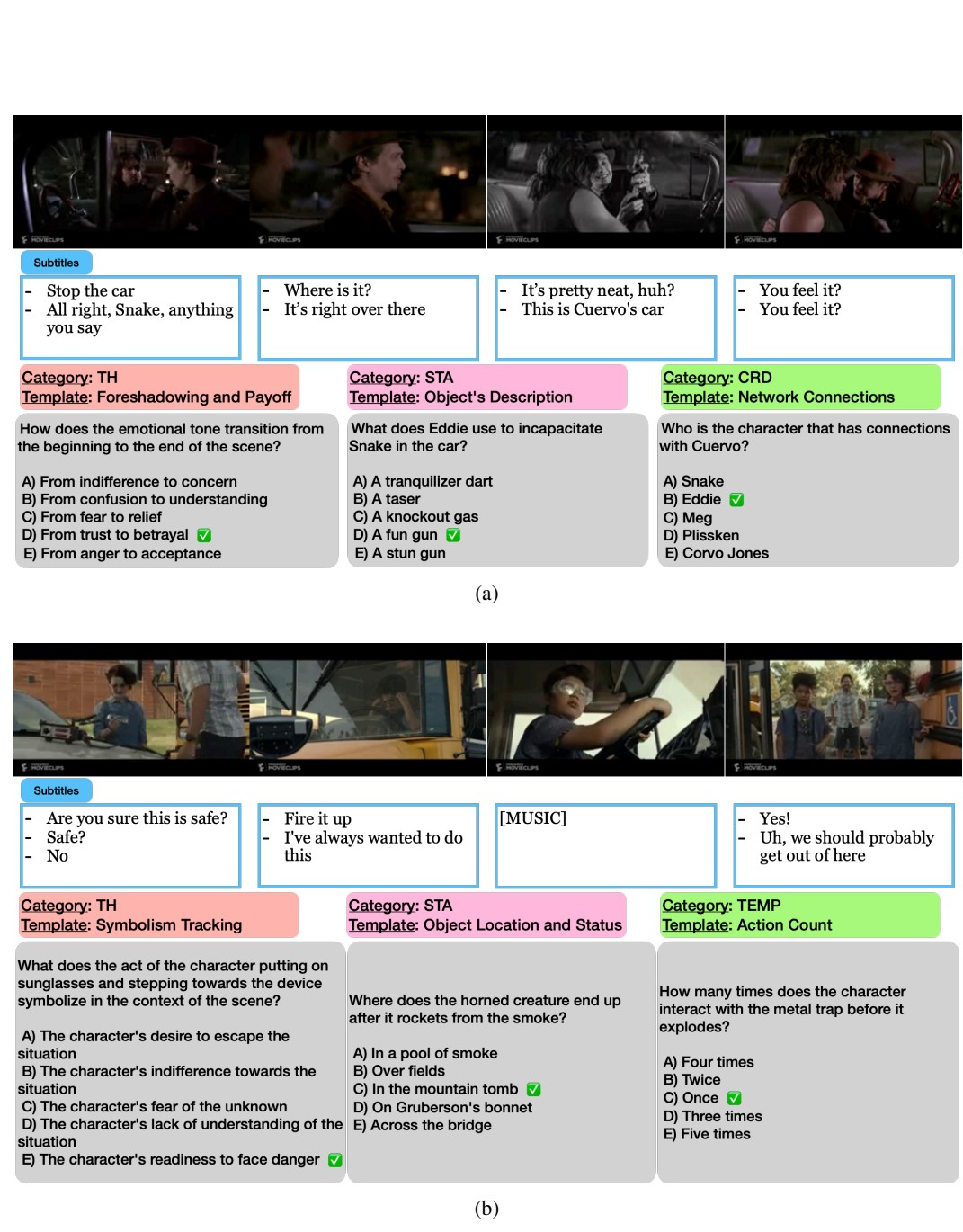

Figure 16: **Example movie clip and multiple-choice questions from CinePile**. The first and second rows depict a selection of image frames extracted from movie clips from (a)Escape From L.A., and (b)Ghostbusters: Afterlife, accompanied by their corresponding subtitles. The next row showcases example questions along with the question template shown in colored headers. TEMP refers to Temporal. Please refer to Table 4 for other acronyms.

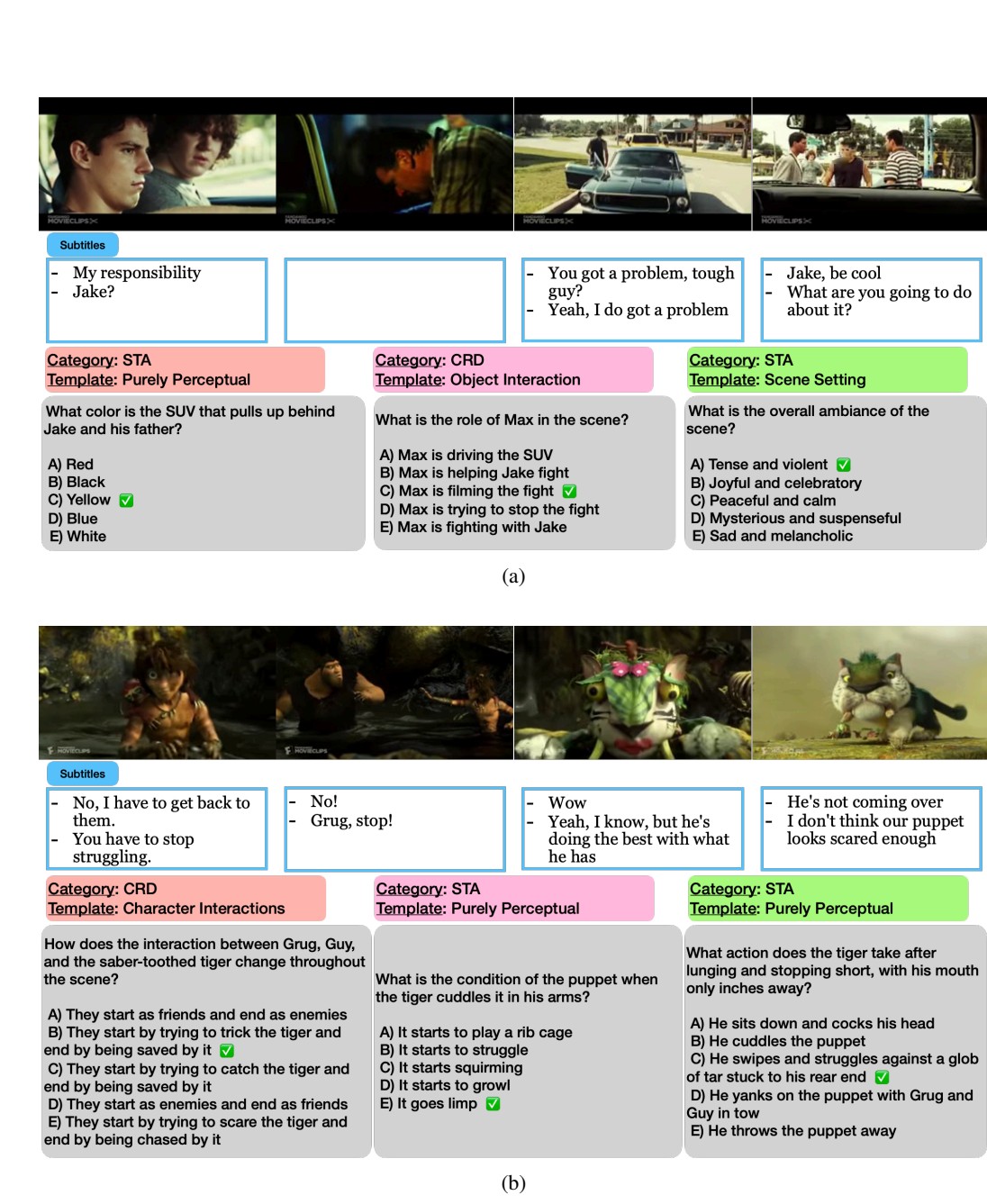

Figure 17: **Example movie clip and multiple-choice questions from CinePile**. The first and second rows depict a selection of image frames extracted from movie clips from (a) Never Back Down, and (b) The Croods, accompanied by their corresponding subtitles. The next row showcases example questions along with the question template shown in colored headers. TEMP refers to Temporal. Please refer to Table 4 for other acronyms.

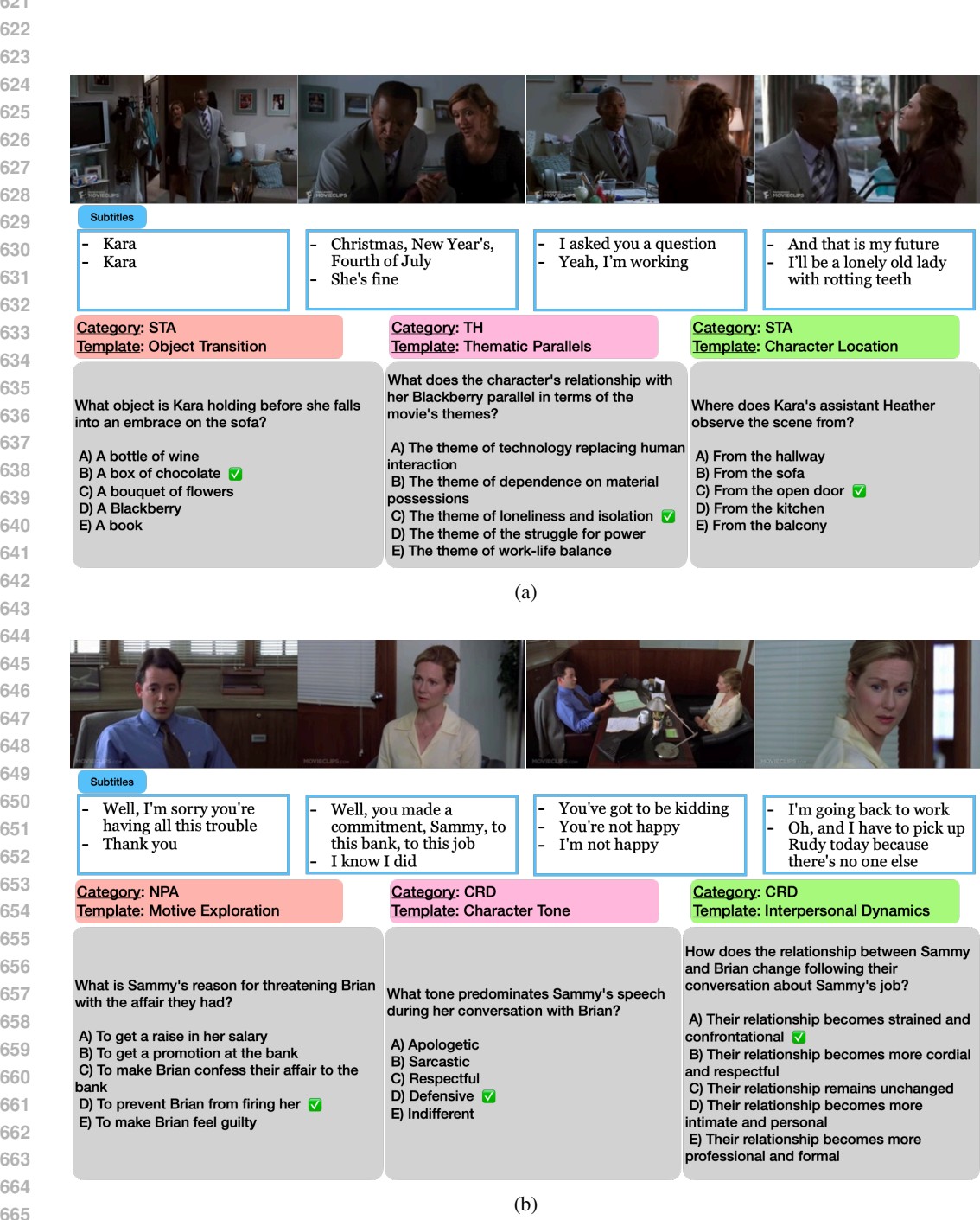

Figure 18: **Example movie clip and multiple-choice questions from CinePile**. The first and second rows depict a selection of image frames extracted from movie clips from (a) Valentine's Day, and (b) You Can Count on Me, accompanied by their corresponding subtitles. The next row showcases example questions along with the question template shown in colored headers. TEMP refers to Temporal. Please refer to Table 4 for other acronyms.

## O  ADAPTING CINEPILE TO LONGER AND DIFFERENT VIDEOS

While we primarily focused on ≈ 160 seconds movie clips as the data source for generating question answers in CinePile, as future models with improved temporal resolution get released, they will require even longer and diverse videos for training and evaluation. To meet this need, CinePile was developed not only as a dataset and benchmark but also as a reproducible, scalable, and efficient pipeline for curating long-form video datasets. In this section, we demonstrate this adaptability by experimenting with three longer videos from diverse domains: Survive 100 Days Trapped, Win $500,000 (1620 seconds, YouTube Challenge-Reward), How Hansi Flick's Tactics Are Revolutionizing Barcelona (540 seconds, soccer tactical analysis), and Eminem - Stan (Long Version) ft. Dido (480 seconds, music video). These videos, vastly different from CinePile's movie clips, were transcribed using Whisper, with key visual descriptions annotated by the authors. Additionally, we slightly revised the question generation prompt to reduce the emphasis on general video analysis (e.g., changing "Create diverse and challenging questions based on the film analysis..." to "Create diverse and challenging questions based on the video analysis..."). We utilized the same question template bank (86 total templates) without adding or removing any. Feeding "video scene information" into our pipeline generated high-quality questions. For instance:

*"What are the strong points of conflict between the characters in the video?"* (video: *Survive 100 Days Trapped, Win $500,000*)

With options:

- *A)* Hot water running out, disinterest in playing board games, rave at 3 a.m.
- *B)* Hot water running out, disinterest in video games, rave at 3 a.m.
- *C)* Essential food running out, hygiene in the bathroom, snoring at night.
- *D)* Essential food running out, disinterest in video games, hygiene in the bathroom.
- *E)* Essential food running out, disinterest in playing board games, hygiene in the bathroom.

Answering this required analyzing the entire clip to identify key conflicts and select the correct option.

Similarly:

*"How does the video develop the theme of Barcelona's tactical variations in attack from start to finish?"* (video: *How Hansi Flick's Tactics Are Revolutionizing Barcelona*)

With options:

- *A)* Dynamic-1: utilizing pace of the attacking wingers, Dynamic-2: slowing the tempo with tiki-taka, Dynamic-3: center-back pinning by the center forward.
- *B)* Dynamic-1: counter-attacks using wingers, Dynamic-2: tiki-taka in possession, Dynamic-3: center forward making constant in-behind runs.
- *C)* Dynamic-1: utilizing the depth created by the full back, Dynamic-2: diagonal runs by the midfielders, Dynamic-3: center-back pinning by the center forward.
- *D)* Dynamic-1: inverted full-backs that come into midfield, Dynamic-2: long balls behind for runs by forwards, Dynamic-3: center defensive midfielder dropping into the backline.
- *E)* Dynamic-1: overlapping full-backs, Dynamic-2: center-back dropping into midfield to push the midfielders up, Dynamic-3: wingers constantly swapping wings to confuse the defense.

Answering this involved identifying and mapping out the tactical variations discussed throughout the video.

These examples demonstrate our pipeline's ability to generalize effectively across different video sources and contexts. Additionally, we evaluated several models on questions generated from these longer videos. The results were as follows: Gemini-Pro-1.5: 41.67% accuracy, GPT-4V: 33.33%, GPT-4o: 41.67%, and LLaVa-OV: 33.33%. This shows that the trend in model performance remains similar; however, as expected, there is a substantial drop in performance compared to the 160-second clips.

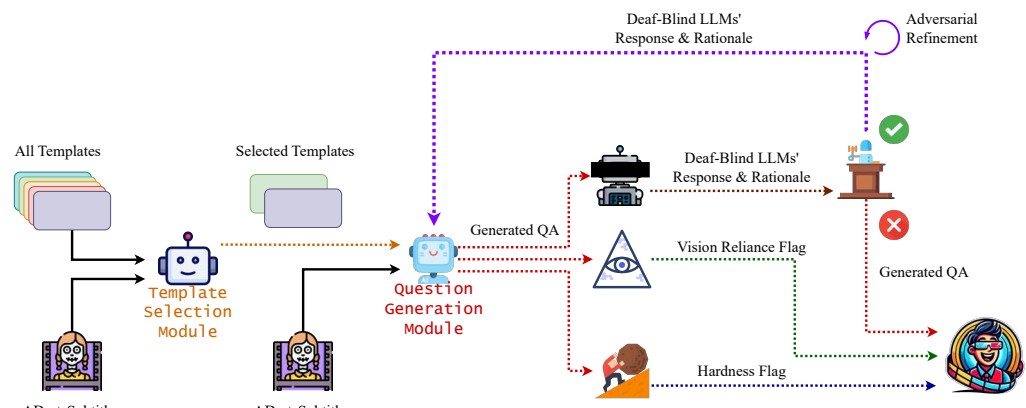

Figure 19: Pipeline demonstrating steps involved in generation, filtration, and refinement of question-answer pairs in CinePile.

# P  ADDITIONAL ADVERSARIAL REFINEMENT DETAILS

**Adjusting for chance performance:** While refining questions in our adversarial refinement pipeline, one concern was that the deaf-blind LLM might only get the right answer by chance. Since our problem involves a multiple-choice QA setup, there is a 25% chance that questions could be answered correctly by a random baseline. Similarly, it was possible that the LLM got the wrong answer due to chance, even though it would be expected to answer correctly the majority of the time. To address this, we devised a methodology where the LLM's response was tested five times using different permutations of the choice order, rotating the options clockwise. We considered the refinement successful only if the LLM failed to answer the question correctly in the majority of cases, i.e., at least three out of five times. If the refinement failed, we repeated the process up to five times, although this is a hyperparameter that can be adjusted based on available computational resources.

**Monetary costs for adversarially refining QAs:** For adversarial refinement, we use GPT-4o for question rephrasing and the free-tier of LLaMA 3.1 70B API provided by Groq. The cost per question fix is only dependent on rephrasing by GPT-4o, and can be calculated as follows:

- Base prompt (instructions for fixing the question): 709 tokens

- Movie scene (subtitles and visual descriptions): 465 tokens (average; varies across scenes)

- Deaf-blind LLM response and rationale: 102 tokens (average; varies across scenes)

- Total Input Tokens per Attempt: 1,276 tokens

- Cost per Input Token (GPT-4o): $2.50 per 1M tokens Input Cost per Attempt: $\frac{1,276}{1,000,000} \times 2.50 = \$0.00319$

- Output Tokens: 74 tokens (average)

- Cost per Output Token: $10.00 per 1M tokens

- Output Cost per Attempt: $\frac{74}{1,000,000} \times 10.00 = \$0.00074$

- Total Cost per Attempt: $\$0.00319 + \$0.00074 = \$0.00393$

- Number of Attempts per Question Fix: Up to 5 (Upper bound, average $\approx$ 3)

- Total Cost per Question Fix: $\$0.00393 \times 5 = \$0.01965$

**Refined QA Examples:** We present a few examples of the weak QAs and the corresponding refined QAs along with the deaf-blind LLM's responses and rationale in Fig. 20.

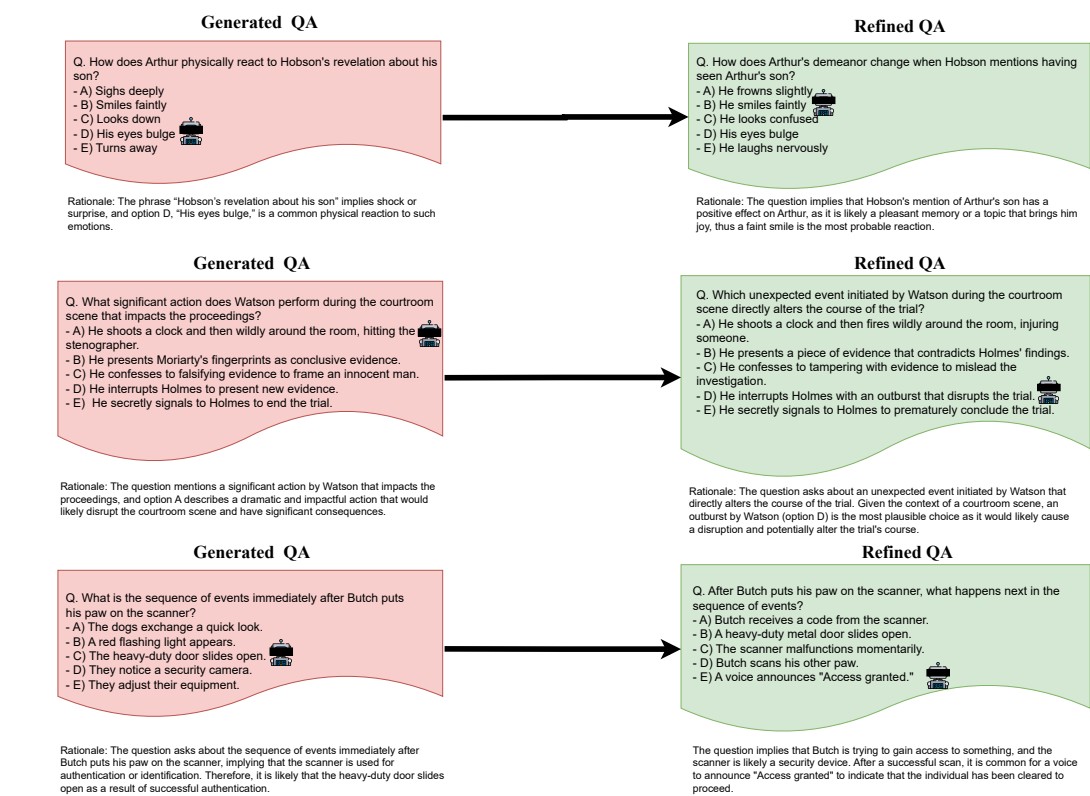

Figure 20: Examples of the weak QAs and the corresponding refined QAs along with the deaf-blind LLM's responses and rationale

# Q  ADDITIONAL DATASET CHARACTERISTICS DETAILS

## Q.1  WITHIN-DATASET ANALYSIS

**Distribution of Dataset Choices.**  One way models can perform well on multiple-choice-based benchmarks is if the correct answer consistently appears in certain positions within the choice order, allowing the model to leverage this information rather than relying on actual understanding. To address this, we randomized all the choices so that the distribution of correct answer positions is approximately uniform. Specifically, the distribution is: "A" (18.72%), "B" (21.35%), "C" (20.18%), "D" (20.26%), and "E" (19.49%), indicating no significant position bias.

**Answer-Distractor Length Similarities.**  Models can perform well on multiple-choice-based benchmarks if the correct answer consistently differs in its linguistic features from the distractor options. For example, the correct answer may often be longer than the distractors. To investigate this, we conducted quantitative experiments analyzing whether the correct option tends to differ in length. Our findings show that the correct answer is the longest option in only 14.18% of the questions, indicating that this occurs in a minority of cases. Similarly, the correct answer is the shortest option in just 5.14% of the questions, demonstrating that no reverse bias exists either. We plot the word count distributions in Fig. 21 for correct answer and distractor options, and in Fig. 22 for the question, correct answer, and different distractor options. We find that, while there is variation across question categories, the answer and distractor options share similar characteristics within each category and, consequently, overall. On average, correct answers have a length of 4.84 words, while distractor options average 4.59 words.

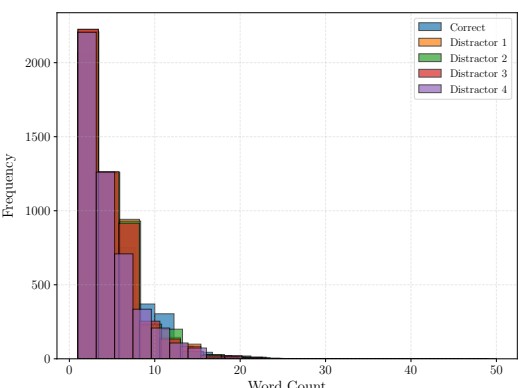

Figure 21: Histograms showing word count distributions for the "correct answer", and the four "distractor" options.

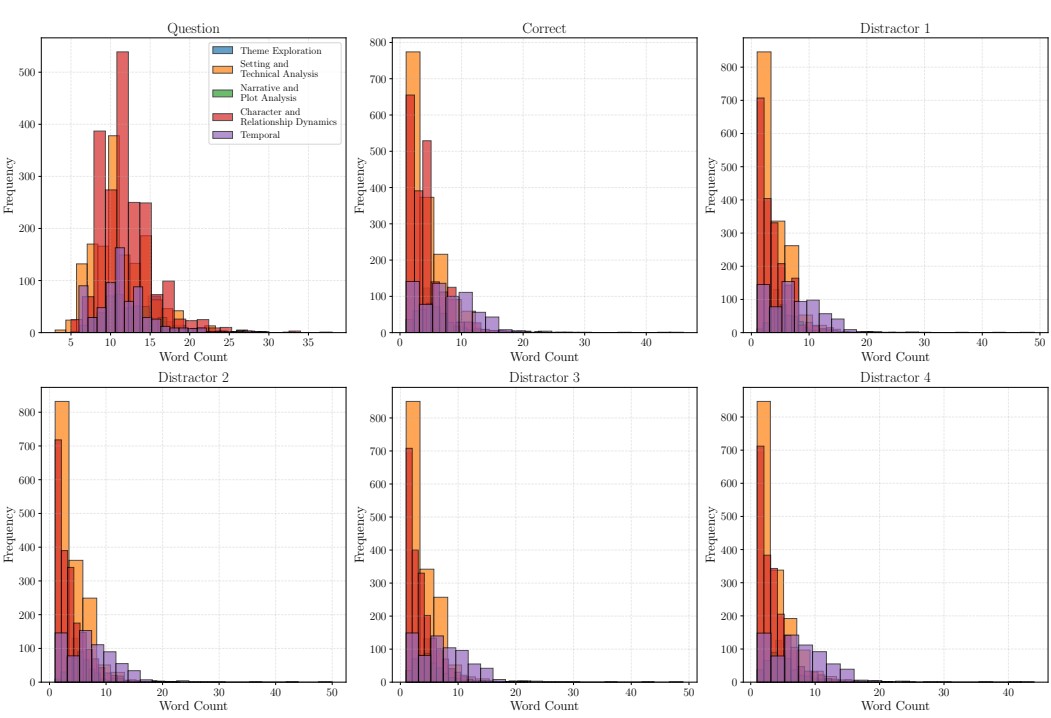

Figure 22: Histograms showing word count distributions for "question", "correct answer", and the four "distractor" options, across different question categories.

## Q.2 COMPARISON WITH OTHER DATASETS

### Q.2.1 QUESTION DIVERSITY

To ensure that the questions in our dataset capture a wide range of aspects, we take the following steps. Firstly, rather than applying fixed templates for every video, we automatically select relevant ones from a diverse bank of 86 templates tailored to various aspects, such as Character Reaction Insight, Event Sequence Ordering, and Moral Dilemma Exploration. Thus, different videos receive different templates, ensuring diversity across the dataset. Secondly, the question generation process is guided by detailed prompts that incorporate both the chosen template and the specific video clip context. As a result, even when the same template is used, the questions vary significantly based on the unique characters, actions, and environments in each video. For example, the questions "How does the decision to buy the coffee machine and the Harry Potter collection lead to a significant consequence in the video?" and "What early tactical trait of Barcelona hinted at their ultimate attacking strategy?" both stem from the "Causal Chain Analysis" template but differ greatly in wording and focus due to the distinct video contexts. This approach contrasts with other datasets relying on human annotators, which often limit template categories (e.g., Perception Test uses four template areas) for human labeling feasibility.

To quantify question diversity, we conducted an experiment to measure the average semantic diversity of questions both within a video clip and across different video clips in our dataset.

**Within-Video Diversity**

For a video clip $v_i$, assume it has $j$ questions $\{q_{i1}, q_{i2}, \ldots, q_{ij}\}$. Using an embedding model, we encoded each question into the embedding space and measured their semantic similarity using cosine similarity $\text{cosim}(q_{ik}, q_{il})$ for all pairs where $1 \leq k, l \leq j$ and $k \neq l$. Since question diversity is inversely related to similarity, we computed the pairwise cosine distance as $1 - \text{cosim}(q_{ik}, q_{il})$. The within-video diversity score for a clip $v_i$ is then given by the expected pairwise cosine distance:

$$D_{\text{within}}(v_i) = \mathbb{E}_{q_{ik}, q_{il} \sim v_i} \left[ 1 - \text{cosim}(q_{ik}, q_{il}) \right]$$

We aggregated this across the dataset by sampling clips $v_i \sim \mathcal{D}$, where $\mathcal{D}$ represents the distribution of video clips in CinePile:

$$D_{\text{within}} = \mathbb{E}_{v_i \sim \mathcal{D}} \left[ D_{\text{within}}(v_i) \right]$$

**Across-Video Diversity:**

To measure diversity across different video clips, we considered the pairwise cosine distances between questions from different videos. For two different video clips $v_i$ and $v_j$ ($i \neq j$), with their associated questions $\{q_{ik}\}$ and $\{q_{jl}\}$, we computed:

$$1 - \text{cosim}(q_{ik}, q_{jl})$$

The across-video diversity score is given by the expected pairwise cosine distance between questions from different videos:

$$D_{\text{across}} = \mathbb{E}_{v_i, v_j \sim \mathcal{D}} \left[ \mathbb{E}_{q_{ik} \sim v_i, q_{jl} \sim v_j} \left[ 1 - \text{cosim}(q_{ik}, q_{jl}) \right] \right], \quad i \neq j$$

**Combined Diversity Score:**

To obtain an overall measure of diversity, we computed the harmonic mean of the within-video and across-video diversity scores:

$$\text{Diversity Score} = 2 \times \frac{D_{\text{within}} \times D_{\text{across}}}{D_{\text{within}} + D_{\text{across}}}$$

The harmonic mean is appropriate in this context because it balances both aspects of diversity by emphasizing the smaller of the two values, and ensuring that neither within-video nor across-video

diversity disproportionately influences the combined score. We compute the diversity score on 50 randomly sampled video clips, and share the results in the table below. CinePile achieves a diversity score of 0.45. For context, we computed the same metric on other datasets: Video-MME: 0.45, MV-Bench 0.42, and IntentQA: 0.37. These comparisons demonstrate the strong semantic diversity of questions in CinePile that is greater or on-par with other (even purely human-curated) datasets.

Table 12: Diversity analysis across datasets based on Within-Video Diversity, Across-Video Diversity, and overall Diversity-Score.

| Dataset | Within-Video Diversity | Across-Video Diversity | Diversity-Score |
|---|---|---|---|
| CinePile | 0.55 | 0.38 | 0.45 |
| Video-MME | 0.53 | 0.40 | 0.45 |
| MVBench | 0.57 | 0.33 | 0.42 |
| IntentQA | 0.45 | 0.32 | 0.37 |

### Q.2.2 MODEL RANKING CORRELATIONS

In this subsection, we compute the Spearman rank correlation ($\rho$) between model ranks on CinePile and their ranks on other datasets, including Video-MME, MV-Bench, and EgoSchema. For each dataset, we use the model ranks provided in their official publications and calculate correlations based on the ranks of models common to both CinePile and the respective dataset. Our results show strong correlations: $\rho = 0.964$ for Video-MME (7 common models, i.e., Gemini 1.5 Pro-001, GPT-4o, Gemini 1.5 Flash-001, GPT-4 Vision, Intern VL-V1.5-25.5, VideoChat2-7B, Video LLaVa-7B), $\rho = 1.000$ for MV-Bench (3 common models, i.e., VideoChat2, Video-ChatGPT-7B, mPLUG-Owl), and $\rho = 1.000$ for EgoSchema (2 common models, i.e., mPLUG-Ow, InternVideo). While CinePile evaluates 26 state-of-the-art models, the number of models evaluated by other benchmarks is often smaller, with limited overlap. For example, MV-Bench assesses only 6 models, of which 3 overlap with CinePile, making some correlations less robust. However, these strong correlations suggest that models performing well on CinePile also perform well on manually curated benchmarks, underscoring CinePile's validity as a reliable test set. That said, performance levels naturally vary due to differences in dataset characteristics and task difficulty. For instance, Gemini-1.5 Pro achieves 81.3% on Video-MME but only 60% on CinePile, highlighting the unique challenges CinePile presents.

## R  OPEN-SOURCE FAILURE MODES

We had previously discussed one of the reasons for why are (some) OSS models so far behind in Sec. 4 of the main paper, where we found that, for extremely poorly performing models (sub 20% overall performance), it was partly due to their inability to follow instructions as we both qualitatively and quantitatively discussed such failure cases in Fig. 9 in the main paper and Appendix Sec. I (Tab. 9). In this section, we discuss a few additional failure modes of open-source models.

**Does Scale (In Parameter Space) Alone Lead to Better Performance?**   There is a lot of focus on model scale these days, so we were curious whether scale alone can lead to better performance (ignoring the architecture, training data, etc). So we computed the Pearson-r correlation between the model scale and overall performance and found it to be weakly positively correlated i.e., 0.157. Obviously, there are alot of confounders across different models like different training data, architecture, etc, so this is not definitely saying that scale would not improve significantly performance, rather it alone is not enough. If we control for everything else by only analyzing one particular model family i.e., InternVL, we see a positive correlation of 0.72.

**Poor ability to utilize visual information; and overdependence on LLM-priors**   Another possible reason for the performance gap in open-source models could be their weaker reliance on visual information and over-reliance on language priors (Tong et al., 2024; Lin et al., 2023). In our experiments (see Appendix Sec. M.1) examining the effect of model performance on the number of sampled frames, we observe that while models improve with additional frames, the extent of this

improvement correlates with the model's overall performance. Specifically, better-performing models tend to utilize visual information more effectively, showing greater performance gains with more frames, whereas weaker models exhibit minimal to no improvement.

**Gap with closed-source models**   The performance advantage of closed-source models likely stems from a combination of factors rather than a single artifact. State-of-the-art models like Gemini-1.5-Pro and GPT-4o operate at scales of hundreds of billions of parameters, significantly outpacing the 7B-26B parameter range of the best open-source models we evaluated. Additionally, while these closed-source models do not disclose details about their training data mixtures or the GPU hours spent, it is reasonable to assume they adhere to scaling laws (Kaplan et al., 2020; Hoffmann et al., 2022) and are trained on datasets that are substantially larger and more diverse than those available to open-source models. The lack of transparency from closed-source models also means there are no ablation studies to pinpoint the optimal combinations of data mixtures or architectural choices contributing to their performance. This makes it challenging to draw precise comparisons.Despite these gaps, open-source models are rapidly catching up, with only about a $\approx$ 10% performance difference in our evaluations. We are optimistic that this gap will continue to shrink in the coming months, and CinePile's training set can be helpful in advancing the capabilities of open-source models.

