# OpenReview forum: "CinePile: A Large-Scale Video Question Answering Dataset and Benchmark"
_ICLR.cc/2026/Conference — ICLR 2026 Conference Withdrawn Submission_

### Official Review · Reviewer_pbBk · 2025-10-30

**Soundness:** 3
**Presentation:** 3
**Contribution:** 2
**Rating:** 4
**Confidence:** 4

**Summary:**

In this work, the authors introduce CinePile, a large-scale dataset and benchmark for long-form video understanding. It consists of 305,000 multiple-choice questions, covering various visual and multimodal aspects. Moreover, they evaluate various video-centric MLLMs and also finetune them for performance investigation.

**Strengths:**

* Clarity

The paper is well-written with good structure. Hence, the clarity is basically good.

* Significance

This paper focuses on evaluating long video understanding of MLLMs, which is an important and practical problem. Hence, the significance is basically OK for video research community.

**Weaknesses:**

Compared to the existing benchmarks, the key contribution of this work is not quite clear. I do appreciate the large scalibility of this dataset. However, the collection process and QA types of this benchmark are basically similar to the exsiting ones.  (Adding Theme type in Table 1  would not change my point too much). Hence, please further clarify the key contributions in this work from these aspects.

**Questions:**

Please see the weakness section.

---

### Official Review · Reviewer_vxTH · 2025-10-30

**Soundness:** 3
**Presentation:** 4
**Contribution:** 3
**Rating:** 6
**Confidence:** 4

**Summary:**

The paper builds a very large long-video QA benchmark, CinePile, by aligning short YouTube movie clips to audio description (AD) transcripts and dialogue, then generating multiple-choice questions (MCQs) with LLMs using a template pipeline distilled from prior, human-curated QA datasets (MovieQA, TVQA, Perception Test). The curation includes relevance filtering, adversarial refinement to reduce degenerate questions, and diagnostics such as “visual-dependence” (answerable from dialogue alone?) and “hardness” (answerable even when the model sees the scene text used for authoring). The result is ~303k QAs across 9,396 clips (avg ≈ 160 s), with ~4,941 MCQs in the test split and a taxonomy spanning character/relations, narrative/plot, temporal, setting/technical, and thematic questions. The authors benchmark 24 proprietary and open-source video-LLMs: humans top out ≈73%, the best proprietary model ≈60%, and the best open-source ≈49%. They also show substantial gains when fine-tuning an open-source baseline (Video-LLaVA) on the CinePile training split.

**Strengths:**

1) The paper presents a scale with valuable utility with 303k QAs (298k train, 5k test) as it fills gap between small benchmarks and provides a training resource that yields 71% relative improvement when fine-tuning Video-LLaVA, proving the dataset's practical value beyond evaluation.

2) The paper presents an automated pipeline that combines audio descriptions as free visual annotations, and template extraction via clustering 30k human questions from MovieQA/TVQA/Perception Test + GPT-4 abstraction into 86 templates.

3) The paper presents a multi-layered quality control with adversarial refinement, by implementing degeneracy filtering using three diverse models (Gemini, GPT-3.5, Phi-1.5), then iteratively repairs ~91% of weak questions using LLaMA-3.1-70B adversarial loop rather than simply discarding them, plus vision-dependence and hardness diagnostics.

4) The paper includes a comprehensive evaluation as the benchmark includes 24 models (proprietary and open-source) with detailed breakdowns across 5 question categories, hard-split analysis showing 15-20% performance drops, frame-rate ablations, and dual human baseline results :73% non-authors, 86% authors.

5) The paper presents a strong case by demonstrating automated generation rivals human-curated diversity despite using templates, by achieving a diversity score of 0.45 using semantic similarity metrics, matching Video-MME and exceeding MVBench (0.42) and IntentQA (0.37),

6) The results show a significant human to model performance gap as it states a 24-point gap between best model (Gemini 1.5 Pro: 60%) and human performance (73%). This indicates that there exists a genuine challenge, with even larger 37-point gap to best open-source model, establishing scope for future progress.

**Weaknesses:**

1) The dataset relies heavily on movie clips with audio descriptions (ADs), and QAs are generated from scene text combining dialogue and visual narration. While this approach seems scalable, it may impose stylistic or narrative biases specific to AD conventions or movie genres. Without a per-source or per-genre performance analysis, it is difficult to assess how such biases shape overall model behavior. The paper reports only high-level category distributions (e.g., 41% “Character & Relationship Dynamics”), leaving this aspect somewhat underexplored.

2) The dataset generation pipeline depends on a small set of LMs (Gemini, GPT-3.5, Phi-1.5), and the “repair” phase uses only LLaMA-3.1-70B. Without any cross-model disagreement analysis, it is unclear whether quality-control decisions might be model-specific. If the authors can add a brief dual-judge consistency check, it could help to strengthen the claims of robustness.

3) Proprietary systems (e.g., Gemini 1.5 Pro) process entire videos, while GPT-4o/4V are restricted to 10-frame samples due to API constraints. As a result, leaderboard comparisons may conflate model competence with input access. A fixed-frames or fixed-token evaluation track would offer a fairer, more interpretable comparison.

4) The introduction of a “hard” test split is one of CinePile’s appealing features—it lowers performance significantly, but the criteria for difficulty labeling are only described qualitatively. If the authors can provide  inter-annotator agreement or per-category reliability for hardness tags, it would make this partition more reproducible and reliable.

5) The visual-dependence analysis usefully distinguishes items answerable from dialogue alone versus those requiring visual cues. However, the finding that many items remain dialogue-solvable suggests that some subsets may still underemphasize video reasoning. If the authors can provide a complementary video-only evaluation, it could help quantify this imbalance.

**Questions:**

1) Would you consider reporting performance and frame-scaling trends by source (MovieClips, AD datasets) and by genre? This would help reveal any residual bias introduced by the AD-driven curation pipeline.

2) In the refinement and filtering stages, have you explored using two independent LMs (e.g., Gemini vs GPT-4) and keeping items on which they disagree? Reporting such disagreement rates would enhance the dataset’s reliability claims.

3) Could you release a small category-wise analysis of distractor types (e.g., plausible vs lexical confounders) and the associated error breakdowns on the “hard” split? This would help determine whether model gains reflect reasoning or test-taking heuristics.

4) What were the instructions, time budgets, and evaluation conditions for “author” and “non-author” human annotators? Reporting confidence intervals or variance across annotators would clarify the significance of the observed gap.

---

### Official Review · Reviewer_SGML · 2025-10-31

**Soundness:** 3
**Presentation:** 2
**Contribution:** 3
**Rating:** 6
**Confidence:** 2

**Summary:**

Authors introduce a large-scale benchmark for long-form video question answering, trying to target actual multimodal and temporal understanding rather than frame-level recognition. It contains ~305K multiple-choice questions across 9,396 movie clips (~160 s each), derived from human audio descriptions of films (for the visually impaired) aligned with video and dialogue.The dataset was built using an LLM-driven automated pipeline.

**Strengths:**

Scalable Dataset Generation Pipeline:
The paper introduces LLM-driven, human-in-the-loop pipeline that uses audio descriptions and automated template generation to create diverse, long-video QA pairs which is scalable, reproducible, and cost-efficient compared to traditional human-only annotation.

Comprehensive Benchmarking and Analysis:
CinePile is evaluated across 24 open-source and proprietary Video-LLMs, with detailed category-wise results (temporal, narrative, thematic, etc.) and human baselines, providing deep diagnostic insight into the current limitations of multimodal reasoning models.

**Weaknesses:**

Limited domain diversity: CinePile relies almost entirely on movie clips, which constrains generalization to real-world, instructional, or egocentric videos where visual and narrative cues differ significantly.


Synthetic question dependency: Despite human-in-the-loop checks, most QAs are LLM-generated, raising concerns about hidden biases, semantic leakage, and superficial reasoning patterns that may not reflect true human-level understanding.

**Questions:**

no much questions at this point.

---

### Official Review · Reviewer_yLyE · 2025-11-11

**Soundness:** 2
**Presentation:** 3
**Contribution:** 2
**Rating:** 2
**Confidence:** 5

**Summary:**

* Introduces CinePile, a large-scale long-form video understanding benchmark: ~305k MCQ QAs across 9,396 videos (train/test).

* Targets temporal reasoning, human–object interactions, and narrative/plot understanding—explicitly aiming to defeat single-frame shortcuts.

* Built via an LLM-driven, human-in-the-loop pipeline that leverages human audio descriptions aligned to public movie clips; questions are generated without direct video input.

* At evaluation time, models get only raw video + dialogue.

* Benchmarks 24 models; humans outperform best commercial models by ~25% and open-source by ~37%; instruction-tuning on CinePile yields large gains (e.g., Video-LLaVA +71% relative).

**Strengths:**

* Clear problem motivation: Existing “long-video” datasets permit frame-based shortcuts; CinePile stresses genuine temporal/narrative comprehension.

* Scale & utility: Large enough for both instruction-tuning and evaluation; already seeing community adoption.

* Generalizable pipeline: Automated QA generation/verification shown to extend to longer (≤30 min) and non-movie videos with minimal prompt changes.

* Question diversity: Broad coverage (temporal, perceptual, reasoning); includes quantitative diversity metrics and a hard split analysis.

**Weaknesses:**

1.	Organization & clarity of exposition
The manuscript would benefit from tighter structure and clearer analysis. For example, Section 3 could be integrated as a subsection of Section 2. Section 2 currently reads as procedural rather than analytical; please quantify key elements (e.g., the proportion of “hard” questions detected and the share of weak Q&A filtered). Statements such as “we observed that some questions…” (L213) should be replaced with concrete statistics and definitions of “trivial/basic.” Reporting these ratios will turn the section from a workflow description into evidence-backed analysis.

2.	Visual reliance policy
If visual grounding is a primary objective, consider filtering out non–vision-centric items rather than merely down-weighting them. Alternatively, justify the scoring approach with ablations showing how the visual-reliance score correlates with model performance and dataset difficulty.

3.	Related work coverage & comparative positioning
The related work and Table 1 under-represent recent benchmarks. Please update with stronger, more current baselines and include a detailed, side-by-side comparison with InfiniBench (released >1 year ago), covering scale, clip duration, domains, question types, and construction pipeline. If CinePile is smaller/shorter, clarify the intended complementary role; if larger/longer in some dimensions, make that explicit.

4.	Template-based generation: motivation & impact
The rationale for using templates needs clarification. What benefits do templates provide over free-form generation (e.g., control, coverage, difficulty)? Quantify any trade-offs: linguistic diversity, semantic variety, and question naturalness. If templates risk constraining the benchmark, consider expanding template sets or adding a free-form subset and report diversity metrics across both.

5.	Evaluation format (MCQ-only)
Limiting the evaluation to MCQ may mask real reasoning ability. Adding an open-ended evaluation regime (short-answer/free-form with calibrated automatic or human grading) would substantially strengthen claims about long-form understanding and reduce the risk of distractor-driven shortcuts.

6.	Input-modality ablations
Please include a systematic analysis of modality contributions: (i) video-only (no subtitles), (ii) subtitles/dialogue-only (no frames), and (iii) full multimodal input. Reporting the deltas will clarify visual vs. textual reliance and help assess potential leakage or superficial cue exploitation.

7.	Baselines and recency in comparisons
Several compared methods in Table 2 appear outdated. Please include major video(-LLM) releases from the past year and ensure consistent, transparent evaluation settings (frame rate, sampling strategy, input modalities, prompt formats) to support fair comparisons.

**Questions:**

1- Compare against InfiniBench in detail, as it seems to offer more skills and longer videos.

2- Please detail all the missing vague numbers in Section 2.

3- Add more recent benchmarks and methods to polish the paper.

4- What are the insights of the benchmark? A beneficial benchmark should shed light on the current limitations to guide future research. Therefore, highlighting interesting conclusions is essential.

---

### Note · Authors · 2025-11-13

I have read and agree with the venue's withdrawal policy on behalf of myself and my co-authors.